# Geospatial Modeling of Health, Socioeconomic, Demographic, and Environmental Factors with COVID-19 Incidence Rate in Arkansas, US

Yaqian He [1,*], Paul J. Seminara [1], Xiao Huang [2], Di Yang [3], Fang Fang [4] and Chao Song [5]

1 Department of Geography, University of Central Arkansas, Conway, AR 72034, USA
2 Department of Geosciences, University of Arkansas, Fayetteville, AR 72701, USA
3 Wyoming Geographic Information Science Center, University of Wyoming, Laramie, WY 82071, USA
4 Department of Urban and Regional Planning, University of Illinois at Urbana-Champaign, Champaign, IL 61820, USA
5 HEOA Group, West China School of Public Health, West China Fourth Hospital, Sichuan University, Chengdu 610044, China
* Correspondence: yhe@uca.edu

**Abstract:** The COVID-19 pandemic has posed numerous challenges to human society. Previous studies explored multiple factors in virus transmission. Yet, their impacts on COVID-19 are not universal and vary across geographical regions. In this study, we thoroughly quantified the spatiotemporal associations of 49 health, socioeconomic, demographic, and environmental factors with COVID-19 at the county level in Arkansas, US. To identify the associations, we applied the ordinary least squares (OLS) linear regression, spatial lag model (SLM), spatial error model (SEM), and multiscale geographically weighted regression (MGWR) model. To reveal how such associations change across different COVID-19 times, we conducted the analyses for each season (i.e., spring, summer, fall, and winter) from 2020 to 2021. We demonstrate that there are different driving factors along with different COVID-19 variants, and their magnitudes change spatiotemporally. However, our results identify that adult obesity has a positive association with the COVID-19 incidence rate over entire Arkansas, thus confirming that people with obesity are vulnerable to COVID-19. Humidity consistently negatively affects COVID-19 across all seasons, denoting that increasing humidity could reduce the risk of COVID-19 infection. In addition, diabetes shows roles in the spread of both early COVID-19 variants and Delta, while humidity plays roles in the spread of Delta and Omicron. Our study highlights the complexity of how multifactor affect COVID-19 in different seasons and counties in Arkansas. These findings are useful for informing local health planning (e.g., vaccine rollout, mask regulation, and testing/tracing) for the residents in Arkansas.

**Keywords:** COVID-19; geographic weighted regression; health; socioeconomic; demographic; environment; Arkansas





## 1. Introduction

### 1.1. Background

The COVID-19 pandemic has been disrupting the lives and livelihoods of people across the world [1–3]. COVID-19 is a respiratory illness caused by the new coronavirus, named severe acute respiratory syndrome coronavirus 2 (SARS-CoV-2) [4]. Since the first case of COVID-19 was reported in December 2019, it has rapidly spread across the globe. According to the World Health Organization [5], the current COVID-19 outbreak has some 545.2 million confirmed cases and 6.33 million deaths (as of 2 July 2022). The consequent public health crisis and associated economic and humanitarian disasters are posing unprecedented impacts on human well-being [6,7]. As such, understanding the driving factors and their associations with COVID-19 transmission is crucial for constraining its spread and promoting future prevention work against similar infectious diseases.

## 1.2. Relevant Research Review

The scientific community has paid enormous efforts to identify the driving factors in COVID-19 virus transmission [8]. The source of infection, the way of transmission, and the susceptible population are the three key elements that decide how an infectious disease spreads [9]. Thus, besides the virus itself, factors related to these elements could play important roles in the spread of COVID-19 [9]. Among multiple potential factors, climatic variables, such as temperature, are the ones researched the most, especially during the initial spread of COVID-19 [10]. For instance, Ganslmeier et al. [11] found that temperature and wind speed have a robust negative impact on the COVID-19 virus spread using ~1.2 million daily observations in nine countries (e.g., Austria, Italy, and the US) for all seasons of 2020. A similar finding was reported by Rosario et al. [12], who stated that high temperature and wind speed likely reduced the spread of COVID-19 in tropical countries. Oppositely, Bashir et al. [13] found a positive correlation between temperature and COVID-19 in New York, US, and Coskun et al. [14] suggested that wind promoted the spread of the COVID-19 virus by increasing air circulation. Precipitation also has a mixed effect on COVID-19 spread. Fern'andez-Ahúja and Martínez [15] revealed that rainfall was not important in explaining the COVID-19 spread in Spain, while Menebo [16] stated that rainfall decreased the spread of virus transmission in Norway by strengthening the 'stay-at-home' order. This mixed effect presented in humidity as well. Wang et al. [17] showed a negative and significant correlation of relative humidity on the reproduction rate of COVID-19 in both the US and China. In contrast, Ahmed et al. [18] claimed that there was very little or nearly no impact of humidity in the outbreak of COVID-19 in 70 cities across the globe.

Climatic variables alone cannot explain all the variability in COVID-19 [19,20]. Several studies have explored the impacts from various non-climatic factors, including socioeconomic, demographic, and health variables, on the virus infection [8,21,22]. Bashir et al. [23] stated that communities with a lower average income in New York City, US, were more at risk of being infected than higher-income communities. Mena et al. [24] drew a similar conclusion regarding Chile, namely that people living in municipalities with a low socioeconomic status did not reduce their mobility during lockdowns as much as those in more affluent municipalities, and thus, they were more vulnerable to COVID-19. However, Yang et al. [19] found that GDP had a positive correlation with COVID-19 in five cities in China. In addition to income-related variables, other non-climatic factors also affect the COVID-19 spread. Da Silva et al. [21] identified the different effects of running water on the COVID-19 spread in different parts of Pernambuco, Brazil, with positive associations in the central region and negative associations in the western and eastern regions.

The above reviews suggest that the impacts of the health, socioeconomic, demographic, and environmental variables on COVID-19 spread differ in different geographic regions, implying that a 'one-size-fits-all' approach may not be appropriate for management and control and indicating the necessity to examine the impacts locally to provide insights on local virus control and prevention. To better understand COVID-19 and its driving factors in the US, considerable efforts have been implemented (Table 1). For example, Igoe et al. [25] identified that COVID-19 hospitalization risks were driven by differences in the socioeconomic, demographic, and health-related factors in the St. Louis area, Missouri, US. Wang et al. [26] explored the associations of city-level walkability, accessibility to biking, public transportation, and socioeconomic factors with COVID-19 cumulative cases in 72 cities in Massachusetts, US. Yet, as one of the states with overwhelming COVID-19 cases, Arkansas received little attention (Table 1). Some studies covering the entire US included the state of Arkansas (Table 1). However, most of these studies only focused on a certain phase of COVID-19 (e.g., February to July 2020 in [22]; 25 January to 29 February 2020 in [19]) and used cumulative cases or deaths over the entire period [26–28]. Nowadays, COVID-19 has evolved into several variants, including Delta and Omicron, which are more contagious. Their driving factors may be different. Qiu et al. [29] found that cities with more medical resources, which were measured by the number of doctors, had lower COVID-19 transmission rates in the early phase of the pandemic (i.e., 19 January to 1 February 2020).

Yet, this effect became insignificant in the second phase of the pandemic (i.e., 2 February to 29 February 2020). Notably, Maiti et al. [30] thoroughly examined the associations of the socioeconomic, health, environmental, demographic, and migration factors with COVID-19 cases and deaths in each month from March to July in the contiguous US and concluded that such associations exhibited temporal variations. Maiti et al. [30] thus recommended that the time dimension needed to be paid more attention to in the spatial epidemiological analysis. However, this study only covered the study period from early to mid-2020, neglecting other COVID-19 variants, mainly because of the availability of COVID-19 data when the study was conducted. Given the inclusive relationships of COVID-19 with different factors across time and space, there is an urgent need to address the effects of different factors on COVID-19 at different time periods in Arkansas locally.

**Table 1.** Previous studies exploring COVID-19 and driving factors in US.

| No. | Reference | Study Focus | Geographic Extent | Study Period | Quantitative Methods |
|---|---|---|---|---|---|
| 1 | Akinwumiju et al. [31] | COVID-19 mortality in relation to the socioeconomic and health conditions | Contiguous US | 1 January–16 September 2020 | Ordinary least squares regression (OLS), spatial lag model (SLM), spatial error model (SME), geographically weighted regression (GWR), and multiscale geographically weighted regression (MGWR) |
| 2 | Ali et al. [27] | COVID-19 occurrence in relation to the socioeconomic, health, and demographic factors | Contiguous US | 1 January–30 June 2020 | Logistic regression |
| 3 | Almalki et al. [32] | COVID-19 cases and death in relation to the socioeconomic and health factors | Guilford County, North Carolina, US | 14 March–14 October 2021 | OLS, GWR, linear multioutput regression, K-nearest neighbors of multioutput regression, random forest of multioutput regression, and support vector regression |
| 4 | Igoe et al. [25] | COVID-19 hospitalization in relation to the socioeconomic, demographic, and health factors | St. Louis Area, Missouri, US | 1 April–30 September 2020 | Univariable global Poisson model and geographically weighted negative binomial (GWNB) model |
| 5 | Iyanda et al. [33] | COVID-19 case fatality ration in relation to the sociodemographic and rural–urban continuum factors | 2407 rural counties, US | 1 January–18 December 2020 | Nonspatial negative binomial Poisson regression and geographically weighted Poisson regression (GWPR) |
| 6 | Juhn et al. [34] | Identify hotspots for COVID-19 | Olmsted County, Minnesota, US | Semimonthly from 11 March–31 October 2020 | Kernel density analysis |
| 7 | Kandula et al. [35] | COVID-19 mortality in relation to the socioeconomic and health factors | US | 1 January–31 December 2020 | Spatial simultaneous autoregressive (SAR) model |
| 8 | Luo et al. [28] | COVID-19 death rate in relation to the socioeconomic, health, environmental, and demographic factors | Contiguous US | 22 January–26 June 2020 | Random forest and geographically weighted random forest (GW-RF) |
| 9 | Lyu et al. [36] | COVID-19 infection rate in relation to the socioeconomic, health, and demographic factors | South Carolina | 7-day window from 1 July–31 December 2020 | SEM, SLM, conditional autoregressive model (CAR), and GWR |
| 10 | Maiti et al. [30] | COVID-19 cases and deaths in relation to the socioeconomic, health, environmental, demographic, and migration factors | Contiguous US | Monthly from 22 January–26 July 2020 | OLS, SEM, SLM, GWR, and MGWR |
| 11 | Mollalo et al. [37] | COVID-19 incidence rate in relation to the socioeconomic and environmental factors | Contiguous US | 22 January–25 April 2020 | Multilayer perceptron neural network |

**Table 1.** *Cont.*

| No. | Reference | Study Focus | Geographic Extent | Study Period | Quantitative Methods |
|---|---|---|---|---|---|
| 12 | Mollalo et al. [38] | COVID-19 incidence rate in relation to the socioeconomic, behavioral, environmental, topographic, and demographic factors | Contiguous US | 22 January–9 April 2020 | OLS, SLM, SEM, GWR, and MGWR |
| 13 | Mollalo et al. [39] | COVID-19 vaccine rate in relation to the socioeconomic and demographic factors | Contiguous US | 11 December, 2020–29 July 2021 | OLS, GWR, and MGWR |
| 14 | Oluyomi et al. [40] | COVID-19 cases in relation to the socioeconomic and health factors | Texas, US | 23 June–3 August 2020 | Poisson regression and GWPR |
| 15 | Tepe [41] | COVID-19 cases in relation to the built and socioeconomic factors | Florida, US | 1 March–23 December 2020 | OLS, SAR, and general spatial model (GSM) |
| 16 | VoPham et al. [42] | COVID-19 cases and mortality in relation to the social distancing, crowding, and socioeconomic factors | US | 18 January–29 April 2020 | Generalized linear mixed model with a Poisson distribution |
| 17 | Wang et al. [26] | COVID-19 rates in relation to the urban environment and socioeconomic factors | 72 cities in Massachusetts, US | 10 April 2021 | Univariate and multivariate regression |
| 18 | Whittle et al. [43] | COVID-19 test positivity rate in relation to the demographic, economic, and health factors | New York, US | 1 March–5 April 2020 | Poisson model with random intercept, Besag–York–Mollié (BYM) model, negative binomial model with random intercept, negative binomial BYM model |
| 19 | Zhang et al. [44] | COVID-19 rate and mortality in relation to the socioeconomic variables | Contiguous US | 18 January–1 May 2020 | Multivariate regression |
| 20 | Wang et al. [45] | COVID-19 case rate and death rate in relation to the socioeconomic and demographic factors | Contiguous US | 18 January–21 July 2020 | Local Spearman's correlation |

*1.3. Methodology Review*

Previous studies have applied different statistical methods to explore the relationship between COVID-19 and various factors. For instance, Wang et al. [45] used a local Spearman's correlation analysis to explore the relationships between COVID-19 cases and the death rate and socioeconomic and demographic factors. Mollalo et al. [37] utilized an artificial neural network to model the COVID-19 incidence rates across the US. Among them, the spatial-related methods were the most used due to their capacity to take spatial autocorrelations into account (Table 1). Combining Bayes smoothing, local Moran's *I*, and bivariate local indicators of spatial association (BiLISA), Mansour et al. [46] identified the spatial associations of the COVID-19 incidence rate with work sectors (e.g., health and agriculture) in Oman. By comparing five different regression models in exploring relationships between COVID-19 mortality and socioeconomic and health conditions, Akinwumiju et al. [31] found the local regression models, named geographically weighted regression (GWR) and multiscale geographically weighted regression (MGWR), outperformed three other global regression models, named ordinary least squares regression (OLS), spatial lag model (SLM), and spatial error model (SME). In addition, Akinwumiju et al. [31] concluded that MGWR was superior to GWR. Similarly, Maiti et al. [30] stated that the performances of GWR and MGWR were better than OLS, SLM, and SME, and MGWR performed better than GWR. This was also confirmed by Mollalo et al. [38] that MGWR could explain the highest variation in the COVID-19 incidence rate compared to OLS, SLM, SEM, and GWR. Due to the effectiveness and wide application of GWR and MGWR in modeling local associations, Comber et al. [47] and Zafri and Khan [48] proposed a route map for using geographically weighted regression reasonably and successfully.

The objective of this study is to quantify the associations of the health, socioeconomic, demographic, and environmental variables with the COVID-19 incidence rate and explore how such associations change across space and time in the state of Arkansas. This study first used a hot spot analysis to identify the spatiotemporal patterns of COVID-19 cases during all seasons from 2020 to 2021, covering the major phases of different COVID-19 variants (i.e., early variant, Delta, and Omicron) in Arkansas. Then, we followed the framework proposed by Comber et al. [47] and Zafri and Khan [48] to apply a OLS, SLM, SEM, and MGWR to model the associations of a total of 49 health, socioeconomic, demographic, and environmental variables with the COVID-19 incidence rate for each season from 2020 to 2021, respectively. The reason that we chose MGWR rather than GWR is shown in Section 2.3.6. This study aimed to answer the following three questions: (1) What are the spatiotemporal patterns of COVID-19 in Arkansas? (2) What are the spatial associations of different driving factors with COVID-19? (3) How do these associations change over time, along with the different COVID-19 variants?

## 2. Materials and Methods

### 2.1. Study Area

The state of Arkansas is located in South-Central US (Figure 1) with a humid subtropical climate. It has an annual precipitation of ~1250 mm and a temperature of ~16 °C, respectively, with relatively hot–humid summers and mild–dry winters [49,50]. Arkansas covers seven ecoregions, including Ozark Highlands, Boston Mountains, Arkansas Valley, Ouachita Mountains, South-Central Plains, Mississippi Alluvial Plain, and Mississippi Valley Loess Plains (Figure 1). Among them, the Mississippi Alluvial Plain, a well-known agricultural region in the US [51], provides abundant productive soil. Agriculture is a key economic driver of Arkansas, contributing more than USD 21 billion annually to its economy [52]. The median household income for Arkansas in 2019 is USD 48,952, USD 16,760 lower than the country's median [53]. Approximately 25.2% of the households have a median income of less than USD 25,000, indicating that Arkansas is an economically stressed state compared to other US states [53]. The life expectancy in Arkansas was only 75.6 years in 2018, ranking 6th lowest among the states and well below the national average of 78.8 years [54]. COVID-19 started to spread in Arkansas on 11 March 2020, and the Delta and Omicron variants started to sweep across the state in June and December 2021, respectively.

### 2.2. Data and Preprocessing

The data on county-level daily cumulative COVID-19 cases from March 2021 to February 2022 were downloaded from the New York Times GitHub database (NYTIMES) (https://github.com/nytimes/covid-19-data (accessed on 1 May 2022)). We collected 24 health, 11 socioeconomic, 9 demographic, and 5 environmental variables from the County Health Rankings (CHR) (https://www.countyhealthrankings.org/ (accessed on 2 May 2022)), the Centers for Disease Control and Prevention (CDC) (https://data.cdc.gov/Vaccinations/COVID-19-Vaccinations-in-the-United-States-County/8xkx-amqh (accessed on 2 May 2022)), the New York Times database, the Agency for Toxic Substances and Disease Registry (ATSDR) (https://www.atsdr.cdc.gov/placeandhealth/svi/data_documentation_download.html (accessed on 1 May 2022)), and the European Center for Medium-Range Weather Forecasts Reanalysis v5 (ERA5) (https://cds.climate.copernicus.eu/#!/search?text=ERA5&type=dataset (accessed on 10 May 2022)). Most variables were yearly data obtained in the years that were close to the study period of 2021 to 2022, but the climate variables, including humidity, precipitation, temperature, and wind, were monthly data. Vaccination variables, including the percentage of persons with at least one dose, percentage of persons aged $\geq$ 5 years with at least one dose, percentage of persons aged $\geq$18 years with at least one dose, and percentage of persons aged $\geq$ 65 years with at least one dose, were daily data. All the variables were at the county scale, except for the climate variables, which were gridded data. The descriptions and sources for each variable are shown in

Table 2. Please note, to be concise, all the factor names in the following context refer to the names in the Column of 'Factors' in Table 2.

**Table 2.** Descriptions and data sources of the health, socioeconomic, demographic, and environmental factors.

| Factors | Descriptions | Sources |
|---|---|---|
| Health | | |
| Poor health | Percentage of adults that report fair or poor health | |
| Poor physical health | Average number of reported physically unhealthy days per month | |
| Poor mental health | Average number of reported mentally unhealthy days per month | |
| Adult smoking | Percentage of adults that reported currently smoking | |
| Adult obesity | Percentage of adults that report Body Mass Index (BMI) >= 30 | |
| Physical inactivity | Percentage of adults that report no leisure-time physical activity | |
| Access to exercise | Percentage of the population with access to places for physical activity | |
| Excessive drinking | Percentage of adults that report excessive drinking | |
| Uninsured | Percentage of people under age 65 without insurance | |
| Primary care physicians rate | Primary care physicians per 100,000 population | CHR |
| Mental health providers rate | Mental health providers per 100,000 population | |
| Flu vaccinations | Percentage of annual Medicare enrollees having an annual flu vaccination | |
| Physical distress | Percentage of adults reporting 14 or more days of poor physical health per month | |
| Mental distress | Percentage of adults reporting 14 or more days of poor mental health per month | |
| Diabetes | Percentage of adults aged 20 and above with diagnosed diabetes (age-adjusted) | |
| Food insecurity | Percentage of population who lack adequate access to food | |
| Limited access to healthy foods | Percentage of population who are low-income and do not live close to a grocery store. | |
| Insufficient sleep | Percentage of adults who report fewer than 7 h of sleep on average (age-adjusted) | |
| % of persons with disability | Percentage of civilian noninstitutionalized population with a disability | ATSDR |
| Mask | The estimated share of people in the county who would always wear masks | NYTIMES |
| % of persons with at least one dose | Percent of the total population with at least one dose of COVID-19 vaccine | |
| % of 5+ persons with at least one dose | Percent of population aged ≥ 5 years with at least one dose of COVID-19 vaccine | CDC |
| % of 18+ persons with at least one dose | Percent of population aged ≥ 18 years with at least one dose of COVID-19 vaccine | |
| % of 65+ persons with at least one dose | Percent of population aged ≥ 65 years with at least one dose of COVID-19 vaccine | |
| Socioeconomic | | |
| High school completion | Percentage of adults aged 25 and over with a high school diploma or equivalent | |
| Some college | Percentage of adults aged 25–44 with some post-secondary education | |
| Unemployment | Percentage of population aged 16+ unemployed and looking for work | |
| Children in poverty | Percentage of children (under age 18) living in poverty | |
| Income inequality | Ratio of household income at the 80th percentile to income at the 20th percentile | CHR |
| Median household income | Median household income ($) | |
| Overcrowding | Percentage of households with overcrowding | |
| Inadequate facilities in house | Percentage of households without of kitchen or plumbing facilities | |
| People in poverty | Percentage of persons below poverty | |
| % of households without vehicle | Percentage of households with no vehicle available estimate | ATSDR |
| % of persons in group quarters | Percentage of persons in group quarters estimate | |
| Demographic | | |
| % of persons below 18 years | Percentage of persons under 18 years of age | |
| % of 65 and older | Percentage of persons older 65 years of age | |
| % of non-Hispanic | Percentage of non-Hispanic Black people | |
| % of native | Percentage of American Indian and Alaska Native people | |
| % of Asian | Percentage of Asian people | CHR |
| % of native Hawaiian | Percentage of native Hawaiian and other Pacific Islander | |
| % of Hispanic | Percentage of Hispanic people | |
| % of non-Hispanic White | Percentage of non-Hispanic White people | |
| % of rural | Percentage of persons living in the rural area | |
| Environmental | | |
| Air pollution | Average daily amount of fine particulate matter ($PM_{2.5}$) in micrograms per cubic meter | CHR |
| Humidity | Specific humidity (kg/kg) | |
| Precipitation | Total precipitation (m) | |
| Temperature | 2-m temperature (°K) | ERA5 |
| Wind | 10-m wind speed (m/s) | |

The county-level COVID-19, mask, and vaccine data covered the entire US. Thus, we first filtered the data only for counties in Arkansas (Figure A1). Similarly, the gridded monthly climate variables were overlapped with the Arkansas county boundaries to extract mean values for each climate variable in each county of Arkansas. Second, to identify how the relationships between COVID-19 and various factors change across different seasons, the daily cumulative COVID-19 cases from March 2020 to February 2021 were calculated for spring (March–April–May: MAM), summer (June–July–August: JJA), fall (September–October–November: SON), and winter (December–January–February: DJF) in 2020 and 2021, respectively. The daily vaccination variables and monthly climate variables were also averaged for each season. All the above processes were conducted using *Python 3.8* libraries, including *pandas*, *xarray*, *rasterio*, and *geopandas*. In doing so, we obtained seasonal COVID-19 cases, health, socioeconomic, demographic, and environmental variables for each county in Arkansas from 2020 to 2021. Note that all the variables with a yearly temporal scale, as

discussed above, remained the same for all seasons (Table 3). The summary of the statistics for each dependent and independent variable is shown in Tables 3 and 4.

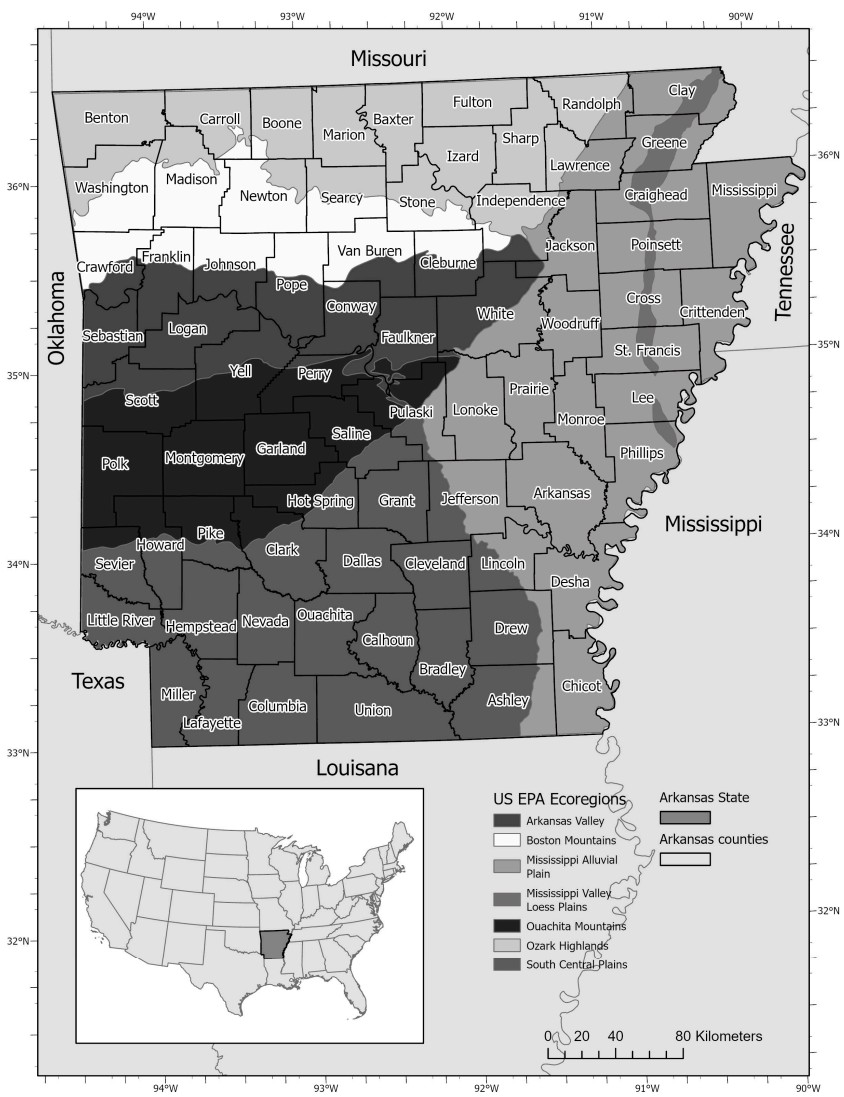

**Figure 1.** The study area of Arkansas.

**Table 3.** Statistical summary for the health, socioeconomic, demographic, and environmental factors with the yearly temporal scale.

| Factors | Mean | STD | MIN | MAX |
|---|---|---|---|---|
| Health | | | | |
| Poor health | 26.4 | 3.1 | 19.0 | 34.0 |
| Poor physical health | 5.4 | 0.4 | 4.2 | 6.2 |
| Poor mental health | 5.3 | 0.3 | 4.4 | 5.9 |
| Adult smoking | 25.1 | 2.2 | 19.0 | 29.0 |
| Adult obesity | 36.4 | 5.6 | 24.0 | 48.0 |
| Physical inactivity | 33.7 | 5.0 | 23.0 | 50.0 |
| Access to exercise | 51.3 | 19.4 | 1.0 | 98.0 |
| Excessive drinking | 15.9 | 1.4 | 12.0 | 19.0 |
| Uninsured | 9.7 | 1.8 | 7.0 | 17.0 |
| Primary care physicians rate | 45.8 | 26.2 | 7.0 | 127.0 |
| Mental health providers rate | 155.1 | 130.4 | 6.0 | 626.0 |
| Flu vaccinations | 42.8 | 8.4 | 21.0 | 56.0 |

**Table 3.** *Cont.*

| Factors | Mean | STD | MIN | MAX |
|---|---|---|---|---|
| Physical distress | 16.6 | 1.4 | 13.0 | 19.0 |
| Mental distress | 17.1 | 1.1 | 14.0 | 19.0 |
| Diabetes | 15.2 | 3.8 | 9.0 | 29.0 |
| Food insecurity | 18.4 | 2.4 | 12.0 | 26.0 |
| Limited access to healthy foods | 10.9 | 7.1 | 1.0 | 33.0 |
| Insufficient sleep | 36.8 | 2.1 | 32.0 | 41.0 |
| % of persons with disability | 20.6 | 3.5 | 9.5 | 27.3 |
| Mask | 0.5 | 0.1 | 0.3 | 0.7 |
| Socioeconomic | | | | |
| High school completion | 84.4 | 3.6 | 76.0 | 92.0 |
| Some college | 52.5 | 7.8 | 33.0 | 69.0 |
| Unemployment | 4.2 | 0.9 | 2.4 | 6.9 |
| Children in poverty | 26.8 | 7.4 | 12.0 | 50.0 |
| Income inequality | 4.8 | 0.7 | 3.7 | 6.8 |
| Median household income | 44,090.4 | 7414.3 | 30,421.0 | 70,775.0 |
| Overcrowding | 2.7 | 1.3 | 1.0 | 7.0 |
| Inadequate facilities in house | 1.2 | 0.8 | 0.0 | 5.0 |
| People in poverty | 19.8 | 4.5 | 8.5 | 33.2 |
| % of households without vehicle | 7.1 | 3.2 | 2.5 | 17.4 |
| % of persons in group quarters | 3.4 | 5.1 | 0.5 | 35.2 |
| Demographic | | | | |
| % of persons below 18 years | 22.2 | 2.4 | 16.3 | 28.6 |
| % of 65 and older | 20.2 | 4.1 | 12.1 | 31.1 |
| % of non-Hispanic | 16.1 | 17.6 | 0.3 | 61.7 |
| % of native | 0.9 | 0.6 | 0.3 | 3.6 |
| % of Asian | 0.9 | 0.8 | 0.1 | 4.7 |
| % of native Hawaiian | 0.2 | 0.4 | 0.0 | 2.8 |
| % of Hispanic | 5.7 | 5.4 | 1.7 | 34.3 |
| % of non-Hispanic White | 74.8 | 17.2 | 34.4 | 94.9 |
| % of rural | 64.9 | 24.1 | 12.3 | 100.0 |
| Environmental | | | | |
| Air pollution | 9.0 | 0.4 | 8.0 | 9.9 |

STD: standard deviation; MIN: minimum value; MAX: maximum value.

Given the potential influence of the county population on COVID-19 cases, we calculated the incidence rate as COVID-19 case number per total population for each county in each season. To directly compare the magnitudes of association between COVID-19 and various factors, we further standardized the 49 factors and COVID-19 rate by subtracting the mean value from each county's value for each factor and then dividing it by the standard deviation, respectively. This process was accomplished using the *Python 3.8 sklearn* library.

**Table 4.** Statistical summary for the COVID-19, health, and environmental factors with the seasonal temporal scale.

| Factors | | 2020 MAM | 2020 JJA | 2020 SON | 2020 DJF | 2021 MAM | 2021 JJA | 2021 SON | 2021 DJF |
|---|---|---|---|---|---|---|---|---|---|
| Cumulative COVID-19 cases | Mean | 94.2 | 710.1 | 1260.6 | 2188.5 | 259.7 | 1462.5 | 1001.1 | 3608.0 |
| | STD | 194.7 | 1179.7 | 1788.9 | 3404.2 | 439.1 | 2242.3 | 1282.1 | 6143.8 |
| | MIN | 0.0 | 24.0 | 126.0 | 247.0 | 14.0 | 120.0 | 73.0 | 327.0 |
| | MAX | 982.0 | 6391.0 | 9833.0 | 20,482.0 | 2432.0 | 14,244.0 | 6274.0 | 41,525.0 |
| Health | | | | | | | | | |
| % of persons with at least one dose | Mean | 0.0 | 0.0 | 0.0 | 4.3 | 24.2 | 35.9 | 46.8 | 52.0 |
| | STD | 0.0 | 0.0 | 0.0 | 1.2 | 4.5 | 6.3 | 7.2 | 7.5 |
| | MIN | 0.0 | 0.0 | 0.0 | 1.1 | 7.1 | 11.3 | 24.2 | 36.3 |
| | MAX | 0.0 | 0.0 | 0.0 | 6.9 | 33.4 | 48.2 | 61.8 | 68.9 |
| % of 5+ persons with at least one dose | Mean | 0.0 | 0.0 | 0.0 | 0.0 | 0.0 | 0.0 | 0.0 | 55.4 |
| | STD | 0.0 | 0.0 | 0.0 | 0.0 | 0.0 | 0.0 | 0.0 | 8.8 |
| | MIN | 0.0 | 0.0 | 0.0 | 0.0 | 0.0 | 0.0 | 0.0 | 34.9 |
| | MAX | 0.0 | 0.0 | 0.0 | 0.0 | 0.0 | 0.0 | 0.0 | 73.4 |
| % of 18+ persons with at least one dose | Mean | 0.0 | 0.0 | 0.0 | 5.5 | 31.0 | 44.3 | 56.3 | 61.7 |
| | STD | 0.0 | 0.0 | 0.0 | 1.5 | 5.8 | 8.1 | 8.9 | 8.9 |
| | MIN | 0.0 | 0.0 | 0.0 | 1.5 | 9.2 | 14.4 | 30.0 | 42.8 |
| | MAX | 0.0 | 0.0 | 0.0 | 9.0 | 44.1 | 61.5 | 76.0 | 82.6 |
| % of 65+ persons with at least one dose | Mean | 0.0 | 0.0 | 0.0 | 11.2 | 56.0 | 65.9 | 73.0 | 78.0 |
| | STD | 0.0 | 0.0 | 0.0 | 3.0 | 9.8 | 10.9 | 10.0 | 9.6 |
| | MIN | 0.0 | 0.0 | 0.0 | 4.2 | 22.0 | 27.0 | 45.1 | 50.9 |
| | MAX | 0.0 | 0.0 | 0.0 | 18.4 | 76.4 | 87.0 | 92.3 | 95.0 |
| Environmental | | | | | | | | | |
| Humidity | Mean | 0.0059 | 0.0105 | 0.0064 | 0.003 | 0.0055 | 0.011 | 0.0063 | 0.0037 |
| | STD | 0.0002 | 0.0001 | 0.0003 | 0.0003 | 0.0003 | 0.0003 | 0.0002 | 0.0003 |
| | MIN | 0.0056 | 0.0102 | 0.0059 | 0.0026 | 0.005 | 0.0101 | 0.0058 | 0.0031 |
| | MAX | 0.0063 | 0.0107 | 0.0068 | 0.0036 | 0.0061 | 0.0115 | 0.0066 | 0.0042 |
| Precipitation | Mean | 0.0052 | 0.0037 | 0.0031 | 0.0038 | 0.0054 | 0.0038 | 0.0024 | 0.0037 |
| | STD | 0.0005 | 0.0005 | 0.0004 | 0.0008 | 0.0006 | 0.0005 | 0.0003 | 0.0006 |
| | MIN | 0.0043 | 0.0028 | 0.0023 | 0.0025 | 0.0043 | 0.0028 | 0.0018 | 0.0025 |
| | MAX | 0.0066 | 0.0051 | 0.0041 | 0.0053 | 0.0075 | 0.0049 | 0.0029 | 0.0051 |
| Temperature | Mean | 289.8 | 299.8 | 290.2 | 278.0 | 289.7 | 300.0 | 291.1 | 281.0 |
| | STD | 1.4 | 0.8 | 1.2 | 1.5 | 1.2 | 0.7 | 0.9 | 1.6 |
| | MIN | 287.1 | 297.7 | 288.0 | 275.5 | 287.1 | 297.9 | 289.2 | 278.3 |
| | MAX | 292.1 | 301.1 | 292.3 | 280.5 | 291.7 | 301.1 | 292.6 | 283.6 |
| Wind | Mean | 3.1 | 2.4 | 2.8 | 3.1 | 3.2 | 2.3 | 2.8 | 3.3 |
| | STD | 0.2 | 0.1 | 0.2 | 0.2 | 0.2 | 0.2 | 0.2 | 0.3 |
| | MIN | 2.6 | 2 | 2.2 | 2.6 | 2.6 | 1.8 | 2.3 | 2.5 |
| | MAX | 3.5 | 2.7 | 3.2 | 3.6 | 3.6 | 2.6 | 3.3 | 3.9 |

STD: standard deviation; MIN: minimum value; MAX: maximum value; MAM: March–April–May; JJA: June–July–August; SON: September–October–November; DJF: December–January–February.

### 2.3. Methods

#### 2.3.1. Health, Socioeconomic, Demographic, and Environmental Variables Selection

As there may be multicollinearity among this large volume of variables, it is necessary to find the most appropriate set of variables for the statistical analysis [55,56]. In this study, we first used the backward stepwise elimination method to select variables for each season from 2020 to 2021 (Figure A1). The backward stepwise elimination method is a widely used variable selection method [57]. It begins with a model that includes all variables and then deletes variables one by one until all remaining variables contribute some significance to the dependent variable [58]. In this study, the contribution was measured by the *p*-value for the *F* statistic that is smaller than the preselected cutoff value (i.e., 0.1 in this study). After backward selection, multicollinearity might still exist, as indicated by large Variance Inflation Factor (VIF) values and Pearson's correlation coefficients (e.g., Table A1 and Figure A2). For any variables with *r* values greater than 0.6 [48], we selected one of them based on the Akaike information criterion (AIC). We also ensured that the selected variables have VIF values under 5. If the selected variables did not contribute statistical significance to the dependent variable, we conducted the backward stepwise elimination method again. Table A2 shows the final selected variables for each model in each season from 2020 to 2021. This process was conducted using the *Python 3.8 statsmodels* and *pandas* libraries.

#### 2.3.2. Hot Spot Analysis

To explore the spatiotemporal patterns of COVID-19 confirmed cases in Arkansas, a hot spot analysis was applied in *ESRI ArcGIS Pro 2.7*. The hot spot analysis can be used to

identify geographic regions with a greater concentration of incidences [59]. The hot spot analysis in *ArcGIS Pro* calculates the Getis-Ord Gi* statistic for each feature (i.e., a county in this study) in a dataset [60]. The Getis-Ord Gi* statistic is expressed as Equation (1) [59].

$$Gi_i^* = \frac{\sum_{j=1}^n w_{i,j} x_j - \overline{x} \sum_{j=1}^n w_{i,j}}{S \sqrt{\frac{n \sum_{j=1}^n w_{i,j}^2 - \left(\sum_{j=1}^n w_{i,j}\right)^2}{n-1}}} \tag{1}$$

where $x_j$ is the attribute value for the feature (i.e., county in this study), $w_{i,j}$ is the spatial weight between features $i$ and $j$, n is the total number of features, $\overline{x}$ is the mean value of $x_j$, and $s = \sqrt{\frac{\sum_{j=1}^n x_j^2}{n} - (\overline{x})^2}$.

The resultant *z*-scores and *p*-values tell where features with either high or low values cluster spatially [61]. A high positive *z*-score with a low *p*-value indicates a cluster of high values (i.e., hot spot), while a low negative *z*-score with a low *p*-value denotes a cluster of low values (i.e., cold spot) [62]. The hot and cold spots highlight the vulnerable and nonvulnerable regions, respectively, of Arkansas during the COVID-19 pandemic [59]. The hot spot analysis for each season was conducted to determine how the patterns of COVID-19 cases change over time.

### 2.3.3. Ordinary Least Squares Linear Regression

According to Comber et al. [47] and Zafri and Khan [48], there are five primary steps that should be undertaken before conducting a GWR variant: (1) a basic linear regression; (2) a spatial autocorrelation and spatial heterogeneity test; (3) a spatial lag model (SLM) or spatial error model (SEM) if spatial autocorrelation was presented; (4) a MGWR if spatial heterogeneity presented; and (5) investigations of the results to decide the appropriate GWR variant (i.e., a standard GWR, a mixed GWR, or a MGWR) (Figure A1). As such, we first undertook an ordinary least squares linear regression model (Equation (2)) for each season from 2020 to 2021 and determined whether there was an autocorrelation and heterogeneity in the residuals.

$$y_i = \beta_0 + \sum_k^n \beta_k x_{ik} + \varepsilon_i \tag{2}$$

where $y_i$ is the dependent variable of the *i*th feature (i.e., the standardized COVID-19 incidence rate for each county in this study), $x_{ik}$ denotes the *k*th independent variables of the *i*th feature (i.e., the standardized variables in Table A2 for each season), $\beta_0$ is the intercept, $\beta_k$ is the regression coefficients, $\varepsilon_i$ is the random error term, and $n$ is the number of independent variables.

To assess the spatial autocorrelation and heterogeneity of the OLS residuals, we calculated the global Moran's *I* and conducted the Breusch–Pagan test for each model, respectively. We also computed *R*-squared, adjusted *R*-squared value, AIC, and corrected AIC (AICc) to evaluate the performance of the OLS models. All processes were conducted using *GeoDa* software.

Section 3.2.1 indicates that SLM, SEM, or GWR variant may be useful, as the Breusch–Pagan test and Moran's *I* values are statistically significant for some seasons. As such, we proceeded with the following steps, i.e., using SLM, SEM, or MGWR to quantify the impacts of the health, socioeconomic, demographic, and environmental variables on COVID-19 cases.

### 2.3.4. Spatial Lag Model

To determine whether to use SLM or SEM if the spatial autocorrelation was represented in the residuals of OLS, we conducted the Lagrange Multiplier (LM)-lag and LM-error tests [48]. For some seasons (e.g., winter 2020), the LM-lag and LM-error are significant, while the robust LM-lag and robust LM-error are insignificant (Table A3), indicating both SLM and SEM are appropriate. Therefore, we built both SLM and SEM and their

performance indicators (i.e., *R*-squared, AIC, Moran's *I*, and Breusch–Pagan test). These processes were conducted using *GeoDa* software.

SLM incorporates spatial autocorrelation between the dependent and independent variables by integrating a spatially lagged dependent variable in the model, which is denoted as [48]:

$$y_i = \beta_0 + \sum_k^n \beta_k x_{ik} + \sum_k^n \rho_k W_i y_i + \varepsilon_i \tag{3}$$

where $y_i$ is the dependent variable of the *i*th feature (i.e., the standardized COVID-19 incidence rate for each county in this study), $x_{ik}$ denotes the *k*th independent variables of the *i*th feature (i.e., the standardized variables in Table A2 for each season), $\beta_0$ is the intercept, $\beta_k$ is the regression coefficients, $\rho_k$ is the spatial autoregressive parameter, $W_i$ is the spatial weights matrix (i.e., the first-order Queens' contiguity weight matrix in this study), $\varepsilon_i$ is the random error term, and $n$ is the number of independent variables.

### 2.3.5. Spatial Error Model

The SEM model treats the error terms of the OLS as spatially correlated [63]. Thus, the error terms are divided into a random error term and a correlated error term [48]. The SEM model is expressed as follows:

$$y_i = \beta_0 + \sum_k^n \beta_k x_{ik} + \sum_k^n \lambda_k W_i \xi_i + \varepsilon_i \tag{4}$$

where $y_i$ is the dependent variable of the *i*th feature (i.e., the standardized COVID-19 incidence rate for each county in this study), $x_{ik}$ denotes the *k*th independent variables of the *i*th feature (i.e., the standardized variables in Table A2 for each season), $\beta_0$ is the intercept, $\beta_k$ is the regression coefficients, $\xi_i$ is the error's spatial component, the intensity of the correlation between these components is the $\lambda_k$, $W_i$ is the spatial weights matrix (i.e., the first-order Queens' contiguity weight matrix in this study), $\varepsilon_i$ is the random error term, and $n$ is the number of independent variables.

### 2.3.6. Multiscale Geographically Weighted Regression Model

The MGWR is an extension of GWR that allows studying the relationships at varying spatial scales by using varying bandwidths as opposed to a single, constant bandwidth used in GWR [64]. The MGWR is expressed as Equation (3) [64].

$$y_i = \beta_0(u_i, v_i) + \sum_k^n \beta_{bwk}(u_i, v_i) x_{ik} + \varepsilon_i \tag{5}$$

where $y_i$ is the dependent variable of the *i*th feature (i.e., the standardized COVID-19 case rate for each county in this study), $x_{ik}$ denotes the independent variables of the *i*th feature (i.e., the standardized variables in Table A2 for each season), $u_i$ and $v_i$ are the spatial coordinates of the *i*th feature (i.e., the centroid coordinates for each county), $\beta_{bwk}(u_i, v_i)$ is the estimated coefficient of the *k*th independent variable for the *i*th feature with the *bw* bandwidth, and $\varepsilon_i$ is the residual at the location $(u_i, v_i)$.

Following Comber et al. [47] and Zafri and Khan [48], we applied a Bi-square local weighting kernel, Golden Search algorithm for bandwidth searching, and corrected Akaike information criterion (AICc) as optimization criteria to develop the MGWR model for each season from 2020 to 2021. The local collinearity in the MGWR model was tested using the condition number [48]. To compare with other models, we also calculated the Moran's *I* for MGWR's residual, *R*-squared value, adjusted *R*-squared value, and AICc for assessing the model performance. All processes were also conducted using *ESRI ArcGIS Pro 2.7*.

As the bandwidths of one or more independent variables deviated from the global bandwidth (Table A4), the MGWR approach would be appropriate for our study [47]. Therefore, we retained the MGWR modeling results to quantify the effects of various factors on COVID-19 in Arkansas for each season from 2020 to 2021.

## 3. Results and Discussion

### 3.1. Spatiotemporal Patterns of COVID-19 Cases in Arkansas

COVID-19 in Arkansas has rapidly increased since it was first reported in March 2020. As of 30 June 2022, Arkansas had 865,592 confirmed COVID-19 cases in total, with 11,581 deaths (https://achi.net/covid19/). COVID-19 in Arkansas has experienced three peaks around January 2021, August 2021, and January 2022, corresponding to the spread of early variants, Delta, and Omicron throughout the state. This results in an increase in the cumulative cases in 2020 DJF, 2021 JJA, and 2021 DJF (Figure 2d,f,h), respectively. Due to the high transmissibility of the Omicron variant, the cumulative COVID-19 cases during 2021 DJF are much higher compared to other seasons (Figure 2h). Throughout all seasons from 2020 to 2021, the COVID-19 cases in Arkansas are mainly distributed in the central, northwestern, and northeastern regions (Figure 2), where Arkansas's three major cities (i.e., Little Rock, Fayetteville, and Jonesboro Lake City) are located.

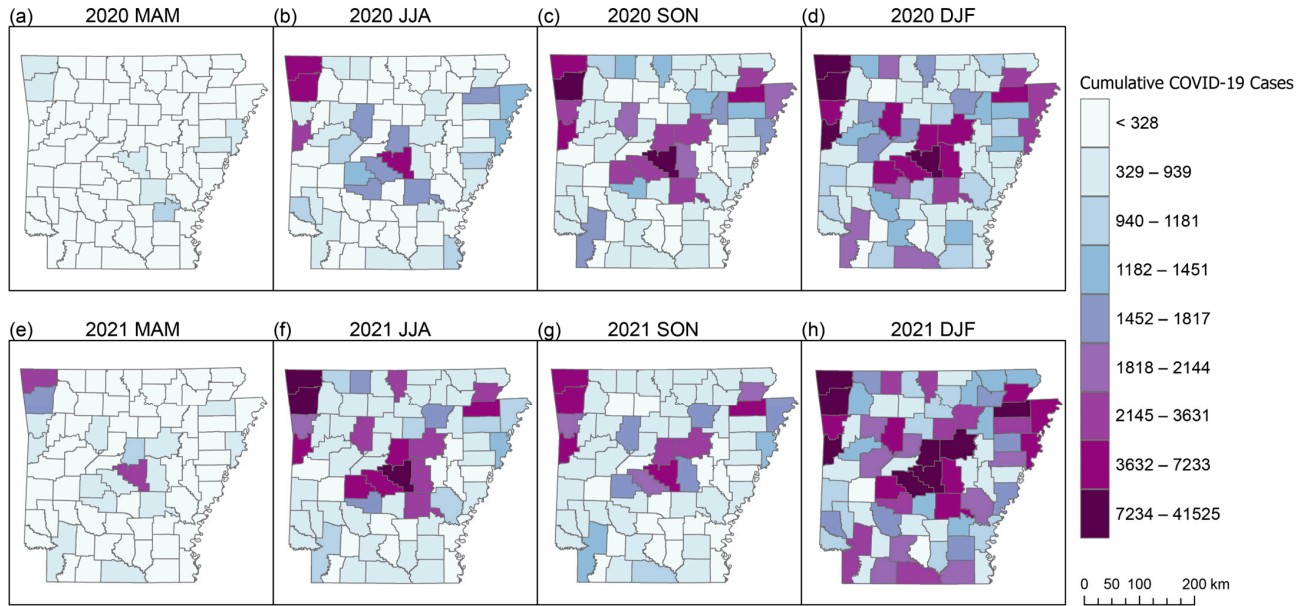

**Figure 2.** The cumulative COVID-19 cases for each season from 2020 to 2021. (MAM: March–April–May; JJA: June–July–August; SON: September–October–November; DJF: December–January–February).

The hot spot analysis further reveals that there are two hot spots in Arkansas (i.e., northwestern and central regions) (Figure 3). This COVID-19 cases' pattern is consistent across all seasons, which further indicates that these regions are more vulnerable to COVID-19 throughout time. This is likely attributed to the high population densities in the regions (Figure A3). A number of studies have concluded that the population size or density was an important factor that influenced the spread of COVID-19. Ahmed et al. [18] found that, for every unit increase in population density (persons per km$^2$), a 14.5% rise in the COVID-19-infected case count could be expected. Yan et al. [65] also claimed that population density was a super factor in increasing the transmission of COVID-19 in the United States. This finding suggested that continuous interventions (e.g., social distancing and quarantine) should be recommended in regions with high population sizes or densities. This notable influence of population density on COVID-19 is the reason for us to use the COVID-19 incidence rate rather than count as the dependent variable, as mentioned in Section 2.2.

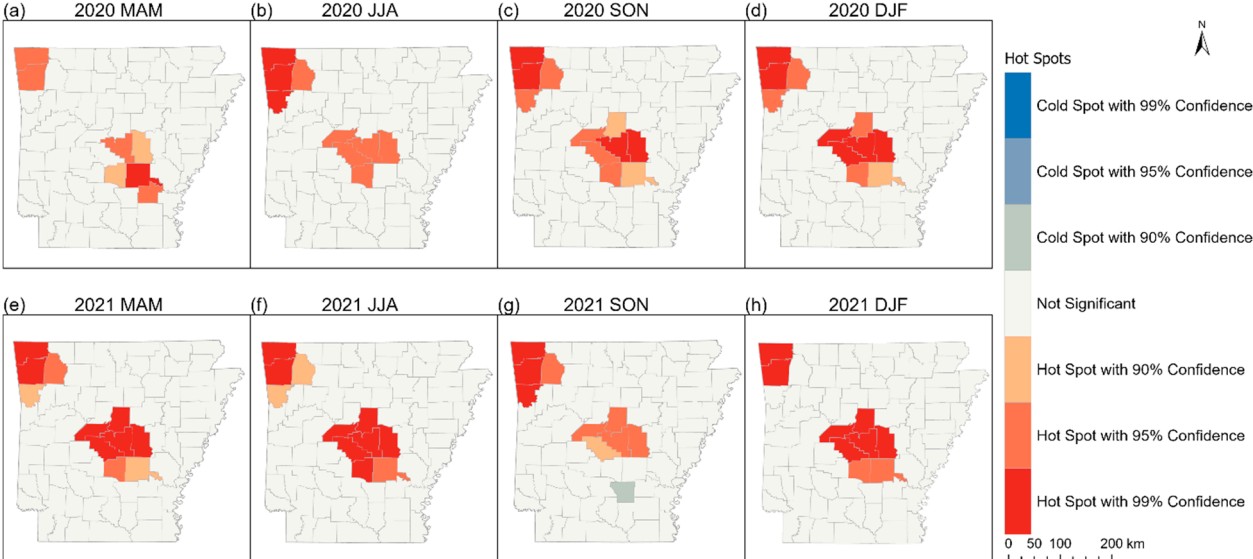

**Figure 3.** Hot spot analysis for the cumulative COVID-19 cases for each season from 2020 to 2021. (MAM: March–April–May; JJA: June–July–August; SON: September–October–November; DJF: December–January–February).

### 3.2. Associations of COVID-19 with Health, Socioeconomic, Demographic, and Environmental Factors

#### 3.2.1. Performance of OLS, SLM, SEM, and MGWR

Table 5 shows the coefficients of the health, socioeconomic, demographic, and environmental factors derived from the OLS, SLM, SEM, and MGWR models, respectively. Table 6 shows their performances. Note, the adjusted *R*-squared value and AICc were not reported by *GeoDa* for both SLM and SEM, and *ArcGIS Pro* did not report the Breusch–Pagan test and AIC for MGWR.

**Table 5.** Coefficients of each health, socioeconomic, demographic, and environmental factor from the OLS, SLM, SEM, and MGWR models for each season from 2020 to 2021.

| | | Coefficient | | | | | |
|---|---|---|---|---|---|---|---|
| | | OLS | SLM | SEM | MGWR | | |
| | | | | | Mean | Min | Max |
| 2020 MAM | Intercept | 0.00 | 0.00 | 0.00 | −0.01 | −0.02 | −0.01 |
| | Adult obesity | 0.31 *** | 0.31 *** | 0.31 *** | 0.32 | 0.28 | 0.36 |
| | Access to exercise | 0.19 ** | 0.19 ** | 0.20 ** | 0.20 | 0.16 | 0.23 |
| | Mental distress | −0.39 *** | −0.39 *** | −0.39 *** | −0.39 | −0.43 | −0.36 |
| | Income inequality | −0.21 ** | −0.21 ** | −0.20 ** | −0.20 | −0.22 | −0.18 |
| | % of persons in group quarters | 0.78 *** | 0.78 *** | 0.79 *** | 0.68 | 0.19 | 1.03 |
| | % of persons below 18 years | 0.24 ** | 0.24 ** | 0.26 *** | 0.27 | 0.25 | 0.29 |
| | %of native | 0.22 ** | 0.22 ** | 0.22 ** | 0.21 | 0.19 | 0.24 |
| | % of rural | 0.49 *** | 0.49 *** | 0.51 *** | 0.47 | 0.46 | 0.48 |
| | Air pollution | −0.22 ** | −0.22 *** | −0.22 *** | −0.22 | −0.23 | −0.21 |
| | Humidity | −0.30 ** | −0.30 *** | −0.30 *** | −0.28 | −0.30 | −0.25 |
| | Temperature | 0.41 *** | 0.41 *** | 0.42 *** | 0.42 | 0.41 | 0.43 |
| 2020 JJA | Intercept | 0.00 | 0.00 | 0.00 | −0.05 | −0.06 | −0.03 |
| | Diabetes | −0.29 *** | −0.29 *** | −0.30 *** | −0.25 | −0.54 | −0.03 |
| | Children in poverty | 0.34 *** | 0.34 *** | 0.38 *** | 0.25 | 0.23 | 0.28 |
| | Income inequality | 0.21 ** | 0.22 ** | 0.20 ** | 0.18 | −0.01 | 0.42 |
| | % of persons in group quarters | 0.36 *** | 0.35 *** | 0.34 *** | 0.34 | 0.32 | 0.37 |
| | % of Hispanic | 0.47 *** | 0.47 *** | 0.49 *** | 0.44 | 0.41 | 0.46 |
| | Humidity | −0.20 ** | −0.20 ** | −0.18 *** | −0.15 | −0.18 | −0.11 |
| 2020 SON | Intercept | 0.00 | 0.00 | −0.01 | −0.05 | −0.26 | 0.31 |
| | Adult obesity | 0.26 ** | 0.19 ** | 0.17 * | 0.16 | 0.11 | 0.20 |
| | % of persons in group quarters | 0.48 *** | 0.48 *** | 0.46 *** | 0.65 | 0.23 | 1.53 |
| | % of non-Hispanic | −0.41 *** | −0.30 *** | −0.19 | −0.32 | −0.58 | −0.13 |
| | Wind | 0.32 *** | 0.16 * | 0.16 | 0.32 | 0.17 | 0.58 |

**Table 5.** *Cont.*

| | | Coefficient | | | | | |
|---|---|---|---|---|---|---|---|
| | | OLS | SLM | SEM | MGWR | | |
| | | | | | Mean | Min | Max |
| 2020 DJF | Intercept | 0.00 | 0.00 | 0.00 | 0.08 | 0.05 | 0.11 |
| | Mental distress | 0.19 * | 0.16 | 0.20 * | 0.17 | 0.16 | 0.19 |
| | Diabetes | 0.30 *** | 0.29 *** | 0.29 *** | 0.34 | 0.14 | 0.40 |
| | % of 18+ persons with at least one dose | 0.35 *** | 0.32 *** | 0.34 *** | 0.30 | 0.18 | 0.49 |
| | % of 65 and older | −0.44 *** | −0.39 *** | −0.42 *** | −0.36 | −0.70 | 0.13 |
| 2021 MAM | Intercept | 0.00 | 0.00 | 0.00 | −0.01 | −0.05 | 0.04 |
| | % of Asian | −0.20 * | −0.18 | −0.19 | −0.22 | −0.27 | −0.18 |
| | % of rural | −0.48 *** | −0.46 *** | −0.48 *** | −0.47 | −0.51 | −0.42 |
| 2021 JJA | Intercept | 0.00 | −0.02 | −0.02 | 0.08 | −0.30 | 0.33 |
| | Poor health | −0.22 * | −0.15 | −0.20 | −0.20 | −0.56 | 0.06 |
| | Access to exercise | −0.31 *** | −0.29 *** | −0.30 *** | −0.33 | −0.36 | −0.29 |
| | Diabetes | 0.29 *** | 0.32 *** | 0.32 *** | 0.30 | 0.26 | 0.34 |
| | Mask | 0.23 ** | 0.20 ** | 0.22 ** | 0.27 | 0.16 | 0.34 |
| | % of 65+ persons with at least one dose | 0.20 ** | 0.22 ** | 0.24 *** | 0.20 | 0.18 | 0.21 |
| | Unemployment | −0.27 ** | −0.28 ** | −0.29 *** | −0.30 | −0.39 | −0.18 |
| | % of non-Hispanic | −0.34 *** | −0.28 ** | −0.25 * | −0.26 | −0.30 | −0.23 |
| | Humidity | −0.23 ** | −0.14 | −0.19 | −0.14 | −0.27 | −0.05 |
| 2021 SON | Intercept | 0.00 | 0.00 | 0.00 | −0.14 | −0.15 | −0.13 |
| | Diabetes | 0.21 ** | 0.19 *** | 0.20 ** | 0.19 | 0.15 | 0.23 |
| | % of persons with at least one dose | −0.24 *** | −0.18 ** | −0.18 ** | −0.18 | −0.45 | 0.03 |
| | Children in poverty | 0.49 *** | 0.43 *** | 0.45 *** | 0.49 | 0.48 | 0.52 |
| | Inadequate facilities in house | −0.20 ** | −0.19 ** | −0.18 ** | −0.19 | −0.21 | −0.17 |
| | % of persons below 18 years | 0.41 *** | 0.36 *** | 0.35 *** | 0.36 | 0.32 | 0.42 |
| | % of Hispanic | 0.32 *** | 0.24 *** | 0.29 *** | 0.30 | 0.29 | 0.31 |
| | % of non-Hispanic White | 0.67 *** | 0.55 *** | 0.62 *** | 0.69 | 0.67 | 0.72 |
| | Humidity | −0.31 *** | −0.17 * | −0.32 *** | −0.27 | −0.49 | 0.05 |
| 2021 DJF | Intercept | 0.00 | 0.00 | 0.00 | 0.09 | −0.26 | 0.37 |
| | % of persons with at least one dose | 0.41 *** | 0.37 *** | 0.41 *** | 0.41 | 0.39 | 0.43 |
| | % of 65 and older | −0.64 *** | −0.57 *** | −0.61 *** | −0.60 | −0.60 | −0.57 |
| | % of Asian | −0.53 *** | −0.44 *** | −0.46 *** | −0.39 | −0.45 | −0.14 |
| | % of native Hawaiian | −0.22 ** | −0.19 ** | −0.19 ** | −0.17 | −0.26 | −0.07 |
| | Humidity | −0.34 *** | −0.31 *** | −0.30 *** | −0.26 | −0.35 | −0.15 |
| | Wind | 0.17 ** | 0.09 | 0.17 * | 0.13 | 0.10 | 0.16 |

*, **, and *** denote the 10%, 5%, and 1% significant levels.

The adjusted *R*-squared value is highest for 2020 MAM (0.68), indicating our selected independent variables could explain 68% of the variation in the COVID-19 incidence rate in the OLS model, whereas the OLS model performed the worst for 2021 MAM, and the selected variables only could explain 17% of the variation in the COVID-19 incidence rate. This is confirmed by the AICc values, ranging from 148.21 for 2020 MAM to 205.36 for 2021 MAM (Table 6). Nevertheless, all OLS models are statistically significant at the 5% level, indicating the reliability of the models.

The Moran's *I* values of the OLS residuals for 2020 SON, 2020 DJF, 2021 JJA, and 2021 SON are statistically significant (Table 6) at the 10% level, indicating that there is spatial autocorrelation in the residuals. This highlights that the OLS approach is not the best approach for modeling the COVID-19 incidence rate in these seasons. Spatial autocorrelation regression modeling (i.e., SLM and SEM) is necessary to improve the performance. Compared to the OLS models, the *R*-squared values for 2020 SON, 2020 DJF, 2021 JJA, and 2021 SON increase from 0.37 to 0.50 (0.49), 0.40 to 0.43 (0.43), 0.45 to 0.52 (0.53), and 0.62 to 0.67 (0.64) for SLM (SEM), respectively. Correspondingly, the AIC values of the OLS models for 2020 SON, 2020 DJF, 2021 JJA, and 2021 SON reduce from 188.06 to 177.44 (178.17), 184.59 to 183.99 (181.93), 186.19 to 181.28 (178.58), 159.10 to 152.43 (156.61) for SLM (SEM), respectively. Moreover, the Moran's *I* values derived from SLM and SEM for those seasons are statistically insignificant (Table 6). All this indicates the effectiveness of SLM and SEM. For 2020 SON and 2021 SON, SLM performs a little better compared to SEM (Table 6). This is consistent with the results of the LM-lag and LM-error tests, which denot that the LM-lag and robust LM-lag tests are statistically significant for 2020 SON and

2021 SON, while the LM-error test and robust LM-error test are insignificant for 2020 SON and 2021 SON, respectively (Table A3). This implies that the SLM is more appropriate for these two seasons, although their differences are minor (Tables 5 and 6).

**Table 6.** Performances of the OLS, SLM, SEM, and MGWR models for each season from 2020 to 2021.

| | | Breusch–Pagan Test | *R*-Squared | Adjusted *R*-Squared | AIC | AICc | Moran's *I* for Residual |
|---|---|---|---|---|---|---|---|
| **2020 MAM** | OLS | 207.06 *** | 0.72 | 0.68 | 140.24 | 148.21 | 0.04 |
| | SLM | 207.06 *** | 0.72 | | 142.24 | | 0.03 |
| | SEM | 204.53 *** | 0.73 | | 140.02 | | 0.00 |
| | MGWR | | 0.82 | 0.77 | | 128.01 | 0.00 |
| **2020 JJA** | OLS | 113.11 *** | 0.56 | 0.52 | 165.44 | 169.62 | −0.09 |
| | SLM | 110.21 *** | 0.56 | | 167.33 | | −0.10 |
| | SEM | 90.56 *** | 0.59 | | 161.93 | | −0.01 |
| | MGWR | | 0.64 | 0.57 | | 167.50 | −0.08 |
| **2020 SON** | OLS | 16.96 *** | 0.37 | 0.34 | 188.06 | 191.29 | 0.20 *** |
| | SLM | 20.17 *** | 0.50 | | 177.44 | | −0.02 |
| | SEM | 22.10 *** | 0.49 | | 178.17 | | 0.01 |
| | MGWR | | 0.58 | 0.52 | | 172.64 | 0.11 * |
| **2020 DJF** | OLS | 0.82 | 0.40 | 0.37 | 184.59 | 187.83 | 0.14 ** |
| | SLM | 1.72 | 0.43 | | 183.99 | | 0.02 |
| | SEM | 1.91 | 0.43 | | 181.93 | | 0.00 |
| | MGWR | | 0.53 | 0.46 | | 181.97 | 0.07 |
| **2021 MAM** | OLS | 1.04 | 0.19 | 0.17 | 202.79 | 205.36 | 0.05 |
| | SLM | 0.96 | 0.20 | | 204.34 | | 0.02 |
| | SEM | 1.07 | 0.20 | | 202.27 | | 0.01 |
| | MGWR | | 0.21 | 0.16 | | 206.05 | 0.05 |
| **2021 JJA** | OLS | 10.59 | 0.45 | 0.38 | 186.19 | 191.63 | 0.19 *** |
| | SLM | 11.32 | 0.52 | | 181.28 | | 0.02 |
| | SEM | 8.3 | 0.53 | | 178.58 | | 0.00 |
| | MGWR | | 0.62 | 0.53 | | 180.26 | 0.06 |
| **2021 SON** | OLS | 5.03 | 0.62 | 0.57 | 159.10 | 164.53 | 0.12 ** |
| | SLM | 4.33 | 0.67 | | 152.43 | | −0.09 |
| | SEM | 5.17 | 0.64 | | 156.61 | | −0.01 |
| | MGWR | | 0.69 | 0.62 | | 160.78 | 0.04 |
| **2021 DJF** | OLS | 1.48 | 0.59 | 0.55 | 160.05 | 164.24 | 0.03 |
| | SLM | 1.54 | 0.61 | | 158.76 | | −0.05 |
| | SEM | 1.42 | 0.59 | | 159.72 | | −0.01 |
| | MGWR | | 0.67 | 0.61 | | 157.68 | −0.05 |

*, **, and *** denote the 10%, 5%, and 1% significant levels.

For the residuals without spatial autocorrelation (i.e., 2020 MAM, 2020 JJA, 2021 MAM, and 2021 DJF), we also built the SLM and SEM for comparison purposes. As expected, the performances of the SLM and SEM and OLS are similar (Table 6). Consistently, the coefficients for different factors also show similar values (Table 5).

The Breusch–Panga tests indicate that there is spatial heterogeneity in the residuals of the OLS models for 2020 MAM, 2020 JJA, and 2020 SON (Table 6). It is necessary to develop MGWR for these seasons. We also still built the models for 2020 DJF, 2021 MAM, 2021 JJA, 2021 SON, and 2021 DJF for comparison purposes and being consistent with the other seasons. The highest local condition number of all the models is 13.5, much smaller than 30, indicating the absence of multicollinearity in the local models. As expected, the adjusted *R*-squared values of the MGWR models for 2020 MAM (0.77), 2020 JJA (0.57), and 2020 SON (0.52) are higher than that of the OLS models (adjusted *R*-squared = 0.68, 0.52, and 0.34, respectively). Consistently, the AICc values of the MGWR models (128.01, 167.50, and 172.64) are lower than that of the OLS models (148.21, 169.62, and 191.29) over the

three seasons (Table 6). Although the residuals of the OLS models for the seasons of 2020 DJF, 2021 JJA, 2021 SON, and 2021 DJF do not have spatial heterogeneity, there are varying bandwidths for the independent variables (Table A4), which may improve the performance of MGWR. However, the adjusted *R*-squared value (0.16) and AICc (206.05) for 2021 MAM of the MGWR model are found to be similar to the OLS model in the same season (adjusted *R*-squared = 0.17 and AICc = 205.36) (Table 6). This is within our expectations, as the residuals of the OLS model for the season are randomly distributed, and the bandwidths are all global values for all independent variables (Table A4).

Collectively, the MGWR models outperform OLS, SLM, and SLM (Table 6). The coefficients for each independent variable for each season have consistent signs for all models (Table 5), and the magnitudes of the coefficients for most of the OLS, SLM, and SEM models are within the range of the MGWR models (Table 5). Thus, in the following discussion, we focus on the MGWR results.

Figure 4 shows the local *R*-squared values of the MGWR models for each county in each season. Some counties in some seasons have low *R*-squared values (e.g., *R*-squared value <0.2 in Benton County in 2020 SON) (Figure 4c). Interestingly, the relatively poor performances of the MGWR models dominate in 2021 MAM (Figure 4e). This may denote that other explanatory variables (e.g., running water [21]) need to be considered in those regions during 2021 MAM. Nevertheless, in most counties and seasons, the local *R*-squared values of the MGWR models are larger than 0.5 (Figure 4), indicating that our models could capture more than 50% of the variability in the standardized COVID-19 case rate. This implies the relative reliability of MGWR in modeling the COVID-19 rate.

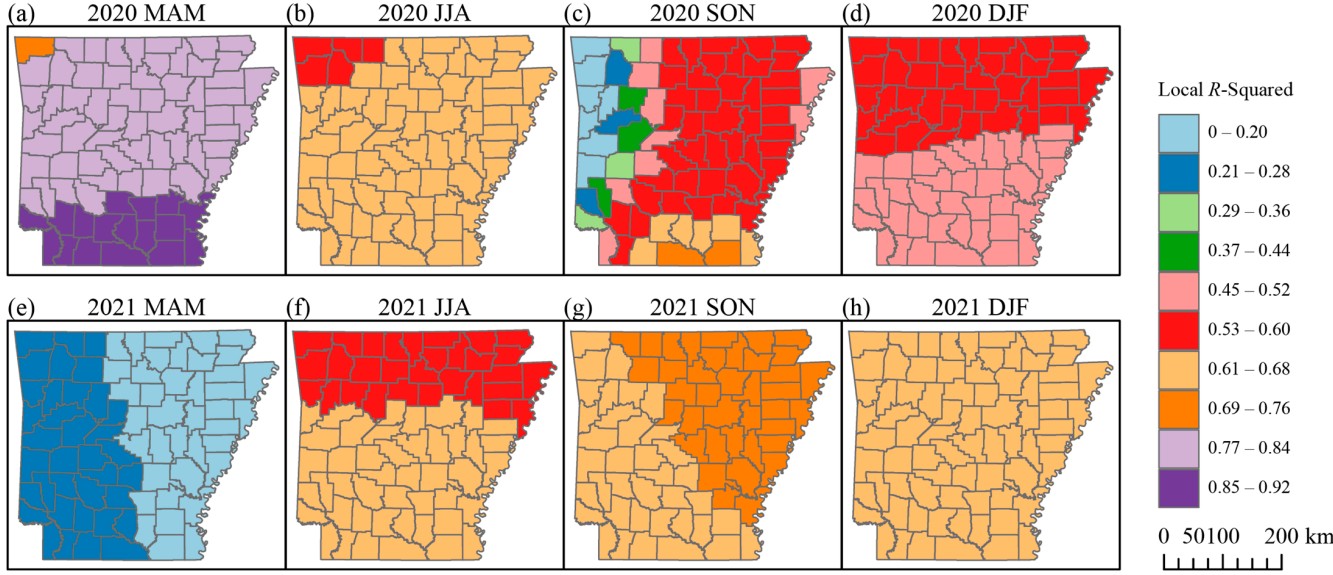

**Figure 4.** The local *R*-squared values for the GWR models in each season from 2020 to 2021. (MAM: March–April–May; JJA: June–July–August; SON: September–October–November; DJF: December–January–February).

### 3.2.2. Associations of COVID-19 with Health Factors

Figures 5–8 present the spatiotemporal impacts of different health, socioeconomic, demographic, and environmental variables on COVID-19 at the Arkansas county level during all seasons from 2020 to 2021. In general, the associations of the variables from the four categories with COVID-19 vary across space and time.

The associations of poor health with COVID-19 are insignificant at the 5% level in most of the counties in Arkansas in the summer of 2021, excepting a few counties in Northern Arkansas, with coefficients ranging from −0.72 to −0.36 (Figure 5a). The significant negative impacts of poor health on COVID-19 may be attributed to people's preferences to stay at home given the fear of COVID-19 [66], thus reducing the risk of exposure. This may

also be the potential reason for the significantly negative associations (coefficients ranging from −0.54 to −0.18) of mental distress with COVID-19 during spring 2020 (Figure 5f).

Adult obesity has significantly positive associations with COVID-19 in all counties during spring 2020, with the highest coefficients (0.36–0.54) in Chicot County (Figure 5b), indicating that people with obesity are vulnerable to COVID-19. Such positive associations persist during fall 2020; however, the effects are not statistically significant at the 5% level (Figure 5c). This finding is consistent with previous studies [67]. Mohammad et al. [68] stated that obesity could weaken the immune system and, therefore, make the host vulnerable to infectious diseases such as COVID-19. Further, Sawadogo et al. [69] demonstrated that people with obesity were at increased risk of both COVID-19-related hospitalizations and death.

The access to exercise shows both significantly negative (coefficients from −0.54 to −0.18) and positive (coefficients from 0.0 to 0.36) associations with COVID-19 during spring 2020 (Figure 5d) and summer 2021 (Figure 5e), respectively. Regular exercise has been shown to improve human immune regulation [70], which may reduce the risk of COVID-19 infection. Yet, exercising may increase the risk of exposure to the virus, especially without social distancing [71]. Arkansas has reopened gyms and fitness centers since 4 May 2020 [72]. As such, people may resume their exercise routines in gyms or fitness centers, which may increase exposure to COVID-19 in a confined environment. However, during the summer, people prefer to exercise outdoors, increasing their contact distance and thus reducing the possibility of COVID-19 infection.

Diabetes shows positive associations with COVID-19 in all counties for winter 2020 (Figure 5i), summer 2021 (Figure 5j), and fall 2021 (Figure 5k), with coefficients ranging from 0 to 0.54, although some counties in Southern Arkansas have insignificant associations (Figure 5i,k)). This is in line with Leon-Abarca et al. [73], who claimed that people with diabetes had a higher probability of being infected after analyzing the cases of more than 1 million Mexican patients. In addition, the positive impacts of diabetes on COVID-19 show spatial variations, with the strongest impacts in Northeastern Arkansas during the winter of 2020 (Figure 5i), while the reason is not clear. Complicating this finding is that diabetes significantly negatively associates with COVID-19 in the eastern part of Arkansas during the summer in 2020 (Figure 5h), with the effects decreasing from the southeastern part (coefficients ranging from −0.72 to −0.54) to northwestern part (coefficients ranging from −0.36 to −0.18). The reasons for the significantly negative associations remain unclear. It may be related to the drugs used by some Type 2 diabetics, as Nyland et al. [74] identified that glucagon-like peptide-1 receptor (GLP-1R) agonists, which have anti-inflammatory effects, were associated with reductions in COVID-19 complications. The stronger negative effects in the southeastern part are probably related to the relatively higher number of people receiving diabetes treatment in this region [75].

As expected, the % of persons with at least one dose shows significantly negative associations with COVID-19 in Eastern Arkansas during fall 2021 (Figure 5m), with coefficients ranging from −0.18 to 0. Moghadas et al. [76] found that vaccination reduced the overall COVID-19 attack rate from 9.0% (without vaccination) to 4.5%. However, vaccination does not always perform as an effective prevention for COVID-19, as stated in Brüssow and Zuber [77]. The % of persons with at least one dose, % of 18+ persons with at least one dose, and % of 65+ persons with at least one dose have significantly positive relationships with COVID-19 during the winter 2020 and 2021 and summer 2021 (Figure 5n–p). This may be due to the fact that people with vaccinations are more likely to break social distances and reduce their willingness to stay at home [78]. Such a reason may also explain the significantly positive associations of masks with COVID-19 during summer 2021 in most Arkansas counties (Figure 5l). The contrasting effects of the % of persons with at least one dose on COVID-19 during fall 2021 (Figure 5m) and winter 2021 (Figure 5n) may be because of changes in people's perceptions. COVID-19 in fall 2021 was at a low level in Arkansas, which may have led people to change their perception of COVID-19 from severe to weak. Therefore, people may lose their COVID-19 practices (e.g., keeping social

distance), resulting in increased infections. There is a spatial variation of the impacts of the % of 18+ persons with at least one dose on COVID-19, with the largest magnitudes in Southern Arkansas, followed by Central Arkansas, in the winter 2020 (Figure 5o). Such variations may be related to the requirement of wearing masks in several universities in Central Arkansas.

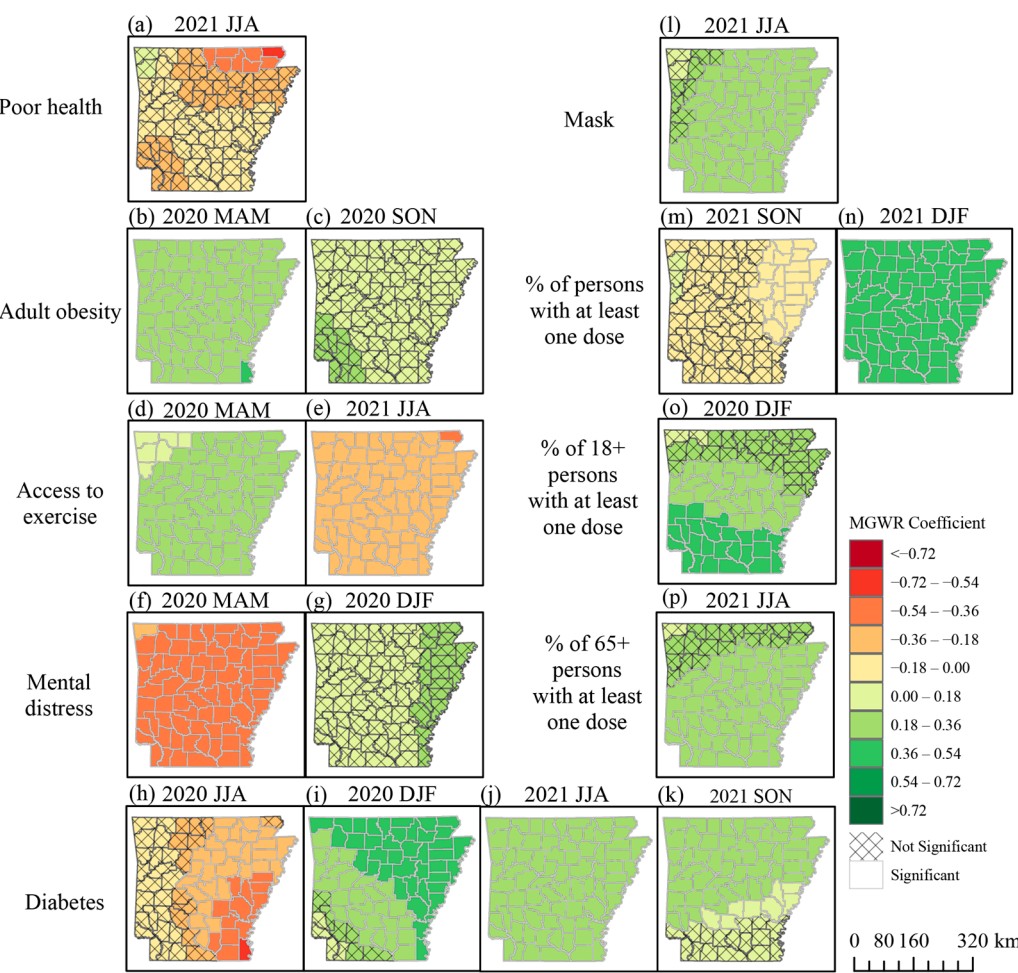

**Figure 5.** Spatial impacts of health variables on the COVID-19 rate derived from a multiscale geographically weighted regression model in each county in Arkansas, US from 2020 to 2021. The unhatched regions indicate a 5% significance level. (MAM: March–April–May; JJA: June–July–August; SON: September–October–November; DJF: December–January–February).

### 3.2.3. Associations of COVID-19 with Socioeconomic Factors

The coefficients of children in poverty with COVID-19 range from 0 to 0.40 in the summer 2020 (Figure 6b) and from 0.40 to 0.80 in the fall 2021 (Figure 6c). Children living in poverty are more likely to experience poor nutrition and live in overcrowded and damp housing and are less likely to have access to green spaces for exercise and less likely to be vaccinated [79]. Thus, they are more vulnerable to catching infectious diseases such as COVID-19.

Consistent with previous studies, income inequality has divergent influences on COVID-19. It has significantly negative effects on the COVID-19 levels in all counties during the spring 2020 (Figure 6d), with the magnitudes ranging from -0.40 to 0, but significantly positive effects (coefficients ranging from 0 to 0.80) on the COVID-19 levels in Eastern Arkansas during the summer 2020 (Figure 6e). Gong and Zhao [80] found that richer people were more mobile and more easily exposed to infection, bringing the virus into their neighborhoods. Oppositely, Demenech et al. [81] found that COVID-19 incidence and mortality increased more pronouncedly among those with greater economic

inequality in Brazilian Federative Units. The different signs of impacts of income inequality on COVID-19 may be due to the fact that richer people are more able to work remotely after recognizing the severity of COVID-19, while poorer people, such as the farmers in Eastern Arkansas, have to go outside to make money [1]. This leads to increased COVID-19 infections.

Unlike Ahmad et al. (2020), who found that poor housing conditions induced higher COVID-19 incidence [82], our results show that inadequate facilities significantly negatively affects the COVID-19 levels in most of the counties in Arkansas during the fall 2021 (Figure 6f). This is probably related to the behavior change that households without a kitchen may directly order ready-to-eat food without visiting the grocery stores because of the fear of COVID-19 [83], thus reducing the infection risk.

In general, the % of persons in group quarters has a significantly positive relationship with the COVID-19 levels (Figure 6g–i), with coefficients ranging from 1.20 to 1.60 in Northern-Central Arkansas during the fall 2020 (Figure 6i) to 0 to 0.40 in all of Arkansas during the summer 2020 (Figure 6h). People living in group quarters (e.g., nursing homes, homeless shelters, dormitories, and prisons) usually share space and facilities (e.g., bathrooms). Thus, such positive relationships are anticipated, as the COVID-19 virus spreads mainly between people who are in close contact with each other [84]. The impacts of the % of persons in group quarters on COVID-19 are strongest in Southern Arkansas, followed by the northern and western parts in the spring 2020 (Figure 6g), while the strongest impacts show in Northern-Central Arkansas, followed by the central and southern parts in the fall 2020 (Figure 6i). These variations may be related to interactions with other factors (Table 5), which needs to be further examined.

Surprisingly, we found negative associations of being unemployment (Figure 6a) with COVID-19 levels. One possible reason is that unemployment people are unlikely to test for COVID-19 given their financial burdens [85], leading to many unreported cases among these groups.

### 3.2.4. Associations of COVID-19 with Demographic Factors

For demographic impacts, there are significantly positive relationships of the % of persons below 18 years (Figure 7a,b), % of native (Figure 7g), % of Hispanic (Figure 7k,l), and % of non-Hispanic White (Figure 7m) with COVID-19 in all counties in Arkansas during different seasons, with coefficients ranging from 0.18 to 0.72. These findings are consistent with previous research. According to the CDC [86], older teens, ages 16 and 17, face the highest rate of weekly cases. This is reinforced by Rumain et al. [87], who concluded that the prevalence of COVID-19 for adolescents (10–19) was significantly greater than that for older adults. McLernon [88] reported that COVID-19 cases, mortality, and case fatality incidences were 2.2, 3.8, and 1.7 times higher for Native Americans compared with White people in Montana. Further, Weeks [89] stated that the American Indian and Alaska Native communities have been experiencing some of the highest rates of COVID-19 in the United States based on the maps generated by The John's Hopkins Coronavirus Resource Center. As Hispanics or Latinos are heavily represented in the service industry, they were 1.5 times more likely to contact COVID-19 than their non-Hispanic White counterparts, as well as 1.9 times more likely to be hospitalized from COVID-19 and 1.8 times more likely to die from COVID-19 [90]. The White population has a relatively low vaccination rate (only 50% of the population are fully vaccinated), leading to a high risk of COVID-19 [91].

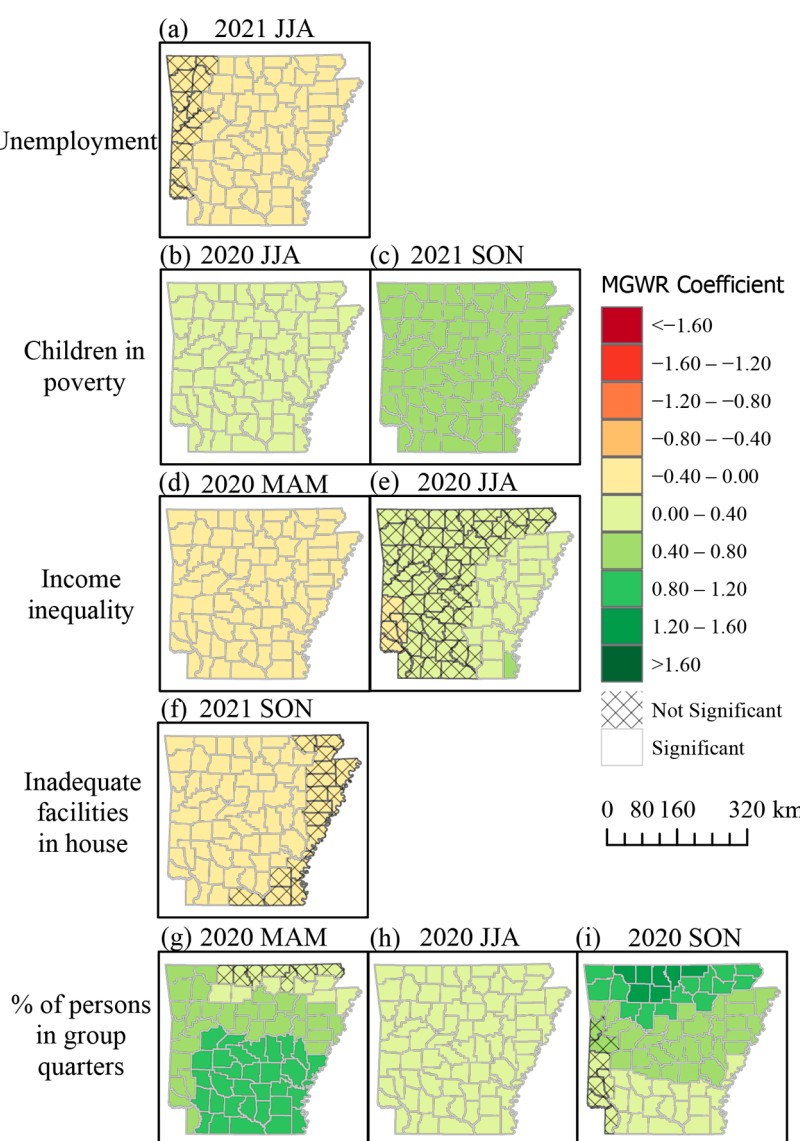

**Figure 6.** Spatial impacts of the socioeconomic variables on the COVID-19 rate derived from the multiscale geographically weighted regression model on COVID-19 in each county in Arkansas, US from 2020 to 2021. The unhatched regions indicate a 5% significance level. (MAM: March–April–May; JJA: June–July–August; SON: September–October–November; DJF: December–January–February).

The % of 65 and older has a significantly negative impact on the COVID-19 levels during the winter 2020 and 2021 (Figure 7c,d), with coefficients ranging from −0.72 to −0.18. Due to the severe illness of COVID-19 in older people, the CDC has suggested giving the top priority to more senior people to get the vaccine [92]. As such, more than 95% of 65 to 74 years old people in the U.S. received at least one dose of the vaccine [91], which thereby may reduce the total COVID-19 burden. In addition, the % of 65 and older shows the strongest negative impacts in Northwestern Arkansas, followed by a belt stretching from the northeastern to southwestern parts in the winter 2020 (Figure 7c). Such variations need to be further explored.

Unlike Native Americans and Hispanics, the % of Asians and % of Native Hawaiians show significantly negative relationships with the COVID-19 levels in Southeastern Arkansas in the spring 2021 (Figure 7h) and Western and Northeastern Arkansas in the winter 2021 (Figure 7i,j). The magnitudes of the relationships range from −0.54 to −0.18. Asians and Native Hawaiian/Pacific Islanders has high vaccination rates. The vaccination may provide some protection for them from COVID-19 [91].

The % of rural parts has both significantly positive (coefficients ranging from 0.36 to 0.54) and negative (coefficients ranging from −0.54 to −0.36) associations with COVID-19 during the spring 2020 (Figure 7n) and spring 2021 (Figure 7o), respectively. The rural area populations tend to be older, sicker, heavier, poorer, and less vaccinated and have experienced higher COVID-19 incidence and mortality rates [93]. Meanwhile, the population density in rural areas is lower compared to metropolitan areas [94], leading to a reduction in COVID-19 transmission. The different signs of impacts are presumably due to the fact that after getting vaccination and a better education, people in rural areas with a lower population density and more open space are less likely to be infected by COVID-19.

As for the impacts of unemployment impacts on COVID-19, the negative influences of the % of non-Hispanic Blacks on COVID-19 (Figure 7e,f) are presumably because of the hidden COVID-19 cases caused by the inability to test for it, which need to be further explored.

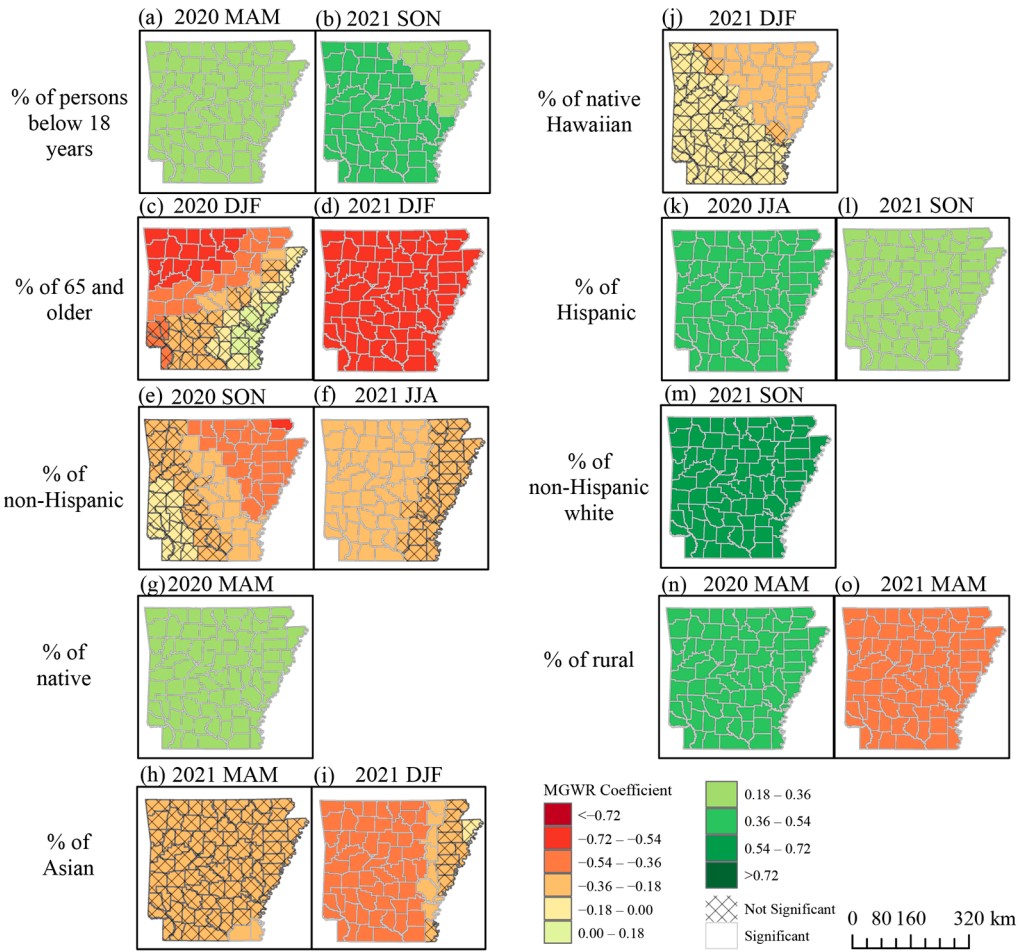

**Figure 7.** Spatial impacts of demographic variables on the COVID-19 rate derived from a multiscale geographically weighted regression model on COVID-19 in each county in Arkansas, US from 2020 to 2021. The unhatched regions indicate a 5% significance level. (MAM: March–April–May; JJA: June–July–August; SON: September–October–November; DJF: December–January–February).

### 3.2.5. Associations of COVID-19 with Climate Factors

Air pollution has significantly negative effects on COVID-19 in all of Arkansas during the spring 2020 (Figure 8a). This supports the previous study that COVID-19 cases were reduced when the amount of $PM_{2.5}$ was above the threshold level, as a higher concentration of $PM_{2.5}$ may restrict human mobility [95].

The humidity consistently shows significantly negative effects on the COVID-19 levels in Arkansas during the spring 2020 (Figure 8b), summer 2021 (Figure 8d), fall 2021 (Figure 8e),

and winter 2021 (Figure 8f). The effects are the strongest in Northeastern Arkansas during the fall 2021, with coefficients ranging from −0.54 to −0.36. These negative effects have been documented by existing studies. For instance, Ward et al. [96] found that a reduction in the relative humidity of 1% was predicted to be associated with an increase of COVID-19 cases by ~6% in New South Wales, Australia. By reviewing 517 articles, Mecenas et al. [97] concluded that a wet climate appeared to reduce the spread of COVID-19. The impacts of humidity on COVID-19 in the fall 2021 show spatial variations, with the largest magnitudes in Northeastern Arkansas, followed by a belt stretching from the northwestern to southeastern parts (Figure 8e). The temperature significantly positively influenced the COVID-19 levels (Figure 8g) in Arkansas during the spring of 2020. Menebo et al. [16] stated that people were more prone to outdoor activities when the sun was shining outside and so eventually became exposed to the virus.

The wind also shows significantly positive impacts on COVID-19 during the fall 2020 (Figure 8h). A high wind speed likely circulated any suspended respiratory droplets in the air and thereby increased the possibility of inhalation by people. The impacts of wind on COVID-19 gradually decrease from Northeastern Arkansas to Southwestern Arkansas (Figure 8h), which is possibly because of the mountainous terrain in Western Arkansas (Figure 1) that blocks the wind to circulate the droplets.

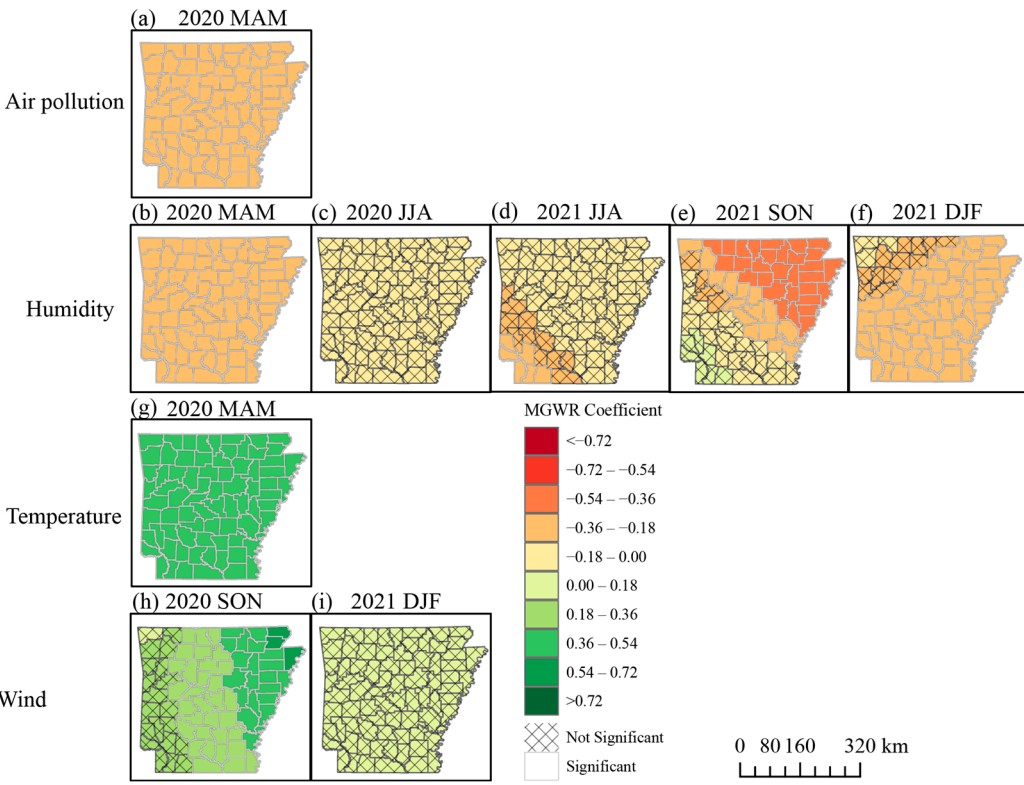

**Figure 8.** Spatial impacts of climate variables on the COVID-19 rate derived from a geographically weighted regression model on COVID-19 in each county in Arkansas, US from 2020 to 2021. The unhatched regions indicate a 5% significance level. (MAM: March–April–May; JJA: June–July–August; SON: September–October–November; DJF: December–January–February).

In summary, different factors contribute to COVID-19 transmission across time. This may be because different variables were involved in the MGWR models in different seasons. For instance, during the spring season in 2020, there were 11 factors from four categories contributing to the COVID-19 rate variation, while, in spring 2021, only two factors from two categories explained the COVID-19 rate variation. Along with different COVID-19 variants (i.e., early variants in winter 2020, Delta in summer 2021, and Omicron in winter

2021), the contributions of driving factors are different. Only diabetes shows consistently positive roles in the spread of both early COVID-19 variants and Delta, while humidity plays consistently negative roles in the spread of both Delta and Omicron.

## 4. Conclusions

The COVID-19 pandemic continues to affect all aspects of society. Attributing the reasons underlying the pandemic's spread at the local scale could help policymakers develop more appropriate prevention strategies to combat COVID-19 and protect local people. In this study, we explored the spatiotemporal patterns of COVID-19 in Arkansas using a hot spot analysis. We then applied a multiscale geographically weighted regression model to thoroughly assess how health, socioeconomic, demographic, and environmental factors affected COVID-19 spread in Arkansas counties, covering all seasons from 2020 to 2021.

The results show that COVID-19 dominates Arkansas' two major cities (Little Rock and Fayetteville). The MGWR models successfully capture the spatial variation of COVID-19 with $R^2$ above 0.5 in most counties across different seasons. The MGWR analyses reveal that people with obesity are more likely to be infected by the COVID-19 virus over all of Arkansas, as well as people with diabetes, despite the existence of negative impacts in mid-2020. Thus, targeted interventions (e.g., distribution of personal protective equipment) should be focused on these groups of people. Exercising generates the potential for people to be more protective against COVID-19. However, our study also reveals that access to exercise positively correlates with COVID-19 cases, suggesting that people should be cautious when exercising, especially to avoid presenting in crowded environments. Consistent with the medical theory, this study finds that vaccination can protect people from COVID-19. Yet, we also reveal that vaccination does not always work. We therefore strongly recommend that vaccinated people still practice social distancing or wearing masks.

Children in poverty and % of persons living in group quarters are both prone to COVID-19 infection. We thus suggest the local government pay great attention to these groups of people, such as distributing more nutrients to children in poverty andincreasing the sanitation conditions in the quarters.

Hispanics are also at risk of COVID-19, largely due to their work sectors (i.e., service industry). Therefore, social distancing and mask are recommended in restaurants or supermarkets, etc. More COVID-19 education (e.g., frequently washing hands and avoiding crowd activities) may be required to reduce the COVID-19 rates in teenagers. We also find that White populations are positively associated with the COVID-19 incidence rate, which may be due to the relative low vaccination rate. Thus, promoting the vaccination rate is still needed for this group of people.

Increased temperature and wind in Arkansas failed to reduce the virus's spread. Thus, we suggest that local people should keep general COVID-19 practices in hot or windy seasons. Humidity shows consistently negative associations with the COVID-19 incidence rate in Arkansas; we thus suggest increasing humidity (e.g., using a humidifier in the bedroom) in residential units.

Our study provides a scientific basis for preventing COVID-19 spread at the county level in Arkansas, US. The methodology used in this study can be applied to any other geographic region that experiences COVID-19 or similar infectious diseases. Yet, there are several limitations in our study. First, there may be other factors influencing COVID-19, such as human mobility and immunity in previous seasons. Future studies need to consider such factors. Second, we find the negative relationships of the unemployment and % of non-Hispanic Black populations with COVID-19, which may be due to hidden COVID-19 cases because of the inability to test it. More accurate data of COVID-19 cases in these two groups are required to better understand their associations with COVID-19. Third, this study did not consider interactions among different factors and interactions between space and time [98,99], and such sophisticated interaction parameters need to be included in the local spatial or spatiotemporal models to improve their performances. Finally, there are

spatial variations in the magnitudes of the effects of different factors on COVID-19. More county-level information, such as education on COVID-19, people's behavior choices, and local terrain information, is needed to better explain these patterns.

**Author Contributions:** Conceptualization, Yaqian He; methodology, Yaqian He, Paul J. Seminara, Xiao Huang, Di Yang, Fang Fang and Chao Song; validation, Yaqian He and Paul J. Seminara; formal analysis, Yaqian He and Paul J. Seminara; data curation, Yaqian He and Paul J. Seminara; writing—original draft preparation, Yaqian He, Xiao Huang, Di Yang, Fang Fang and Chao Song; writing—review and editing, Yaqian He, Xiao Huang, Di Yang, Fang Fang and Chao Song; visualization, Yaqian He, Xiao Huang, Di Yang, Fang Fang and Chao Song; supervision, Yaqian He; and funding acquisition, Yaqian He and Paul J. Seminara. All authors have read and agreed to the published version of the manuscript.

**Funding:** This study was supported by the Arkansas INBRE grant (# 5P20 GM103429) to Yaqian He and the Advancement of Undergraduate Research in the Sciences (AURS) Research Award at the University of Central Arkansas to Paul J. Seminara. Chao Song was supported by National Natural Science Foundation of China (# 42071379).

**Data Availability Statement:** The daily cumulative COVID-19 cases are available in the New York Times GitHub database (https://github.com/nytimes/covid-19-data (accessed on 1 May,2022)). The health, socioeconomic, demographic, and environmental variables are from the County Health Rankings (https://www.countyhealthrankings.org/ (accessed on 2 May 2022)), the Centers for Disease Control and Prevention (https://data.cdc.gov/Vaccinations/COVID-19-Vaccinations-in-the-United-States-County/8xkx-amqh (accessed on 2 May 2022)), the New York Times database, the Agency for Toxic Substances and Disease Registry (https://www.atsdr.cdc.gov/placeandhealth/svi/data_documentation_download.html (accessed on 1 May 2022)), and the European Center for Medium-Range Weather Forecasts Reanalysis v5 (ERA5) (https://cds.climate.copernicus.eu/#!/search?text=ERA5&type=dataset (accessed on 10 May 2022)).

**Acknowledgments:** We are grateful to Jerry Ware and Eric R. Siegel from the Department of Biostatistics at the University of Arkansas for Medical Sciences for support and proofreading our manuscript. We are also grateful to the Reviewers for the constructive comments for improving the manuscript.

**Conflicts of Interest:** The authors declare no conflict of interest.

## Appendix A

**Table A1.** VIF values for different variables in 2021 DJF after backward selection.

| Factors | VIF |
|---|---|
| Adult smoking | 10.46 |
| Excessive drinking | 4.62 |
| Physical distress | 17.19 |
| Food insecurity | 15.09 |
| Insufficient sleep | 6.46 |
| People in poverty | 10.00 |
| % of 65 and older | 1.97 |
| % of non-Hispanic | 2534.19 |
| % of native | 8.22 |
| % of Asian | 9.98 |
| % of native Hawaiian | 3.85 |
| % of Hispanic | 209.35 |
| % of non-Hispanic White | 2408.24 |
| Humidity | 2.96 |
| Wind | 1.89 |
| % of persons with at least one dose | 1.86 |

**Table A2.** Selected variables for each model in each season.

| Factors | Seasons | | | | | | | |
|---|---|---|---|---|---|---|---|---|
| | **2020 MAM** | **2020 JJA** | **2020 SON** | **2020 DJF** | **2021 MAM** | **2021 JJA** | **2021 SON** | **2021 DJF** |
| Health | | | | | | | | |
| Poor health | | | | | | √ | | |
| Poor physical health | | | | | | | | |
| Poor mental health | | | | | | | | |
| Adult smoking | | | | | | | | |
| Adult obesity | √ | | √ | | | | | |
| Physical inactivity | | | | | | | | |
| Access to exercise | √ | | | | | √ | | |
| Excessive drinking | | | | | | | | |
| Uninsured | | | | | | | | |
| Primary care physicians rate | | | | | | | | |
| Mental health providers rate | | | | | | | | |
| Flu vaccinations | | | | | | | | |
| Physical distress | | | | | | | | |
| Mental distress | √ | | | √ | | | | |
| Diabetes | | √ | | √ | | √ | √ | |
| Food insecurity | | | | | | | | |
| Limited access to healthy foods | | | | | | | | |
| Insufficient sleep | | | | | | | | |
| % of persons with disability | | | | | | | | |
| Mask | | | | | | √ | | |
| % of persons with at least one dose | | | | | | | √ | √ |
| % of 5+ persons with at least one dose | | | | | | | | |
| % of 18+ persons with at least one dose | | | | √ | | | | |
| % of 65+ persons with at least one dose | | | | | | √ | | |
| Socioeconomic | | | | | | | | |
| High school completion | | | | | | | | |
| Some college | | | | | | | | |
| Unemployment | | | | | | √ | | |
| Children in poverty | | √ | | | | | √ | |
| Income inequality | √ | √ | | | | | | |
| Median household income | | | | | | | | |
| Overcrowding | | | | | | | | |
| Inadequate facilities in house | | | | | | | √ | |
| People in poverty | | | | | | | | |
| % of households without vehicle | | | | | | | | |
| % of persons in group quarters | √ | √ | √ | | | | | |
| Demographic | | | | | | | | |
| % of persons below 18 years | √ | | | | | | √ | |
| % of 65 and older | | | | √ | | | | √ |
| % of non-Hispanic | | | √ | | | √ | | |
| % of native | √ | | | | | | | |
| % of Asian | | | | | √ | | | √ |
| % of native Hawaiian | | | | | | | | √ |
| % of Hispanic | | √ | | | | | √ | |
| % of non-Hispanic White | | | | | | | √ | |
| % of rural | √ | | | | √ | | | |
| Environmental | | | | | | | | |
| Air pollution | √ | | | | | | | |
| Humidity | √ | √ | | | | √ | √ | √ |
| Precipitation | | | | | | | | |
| Temperature | √ | | | | | | | |
| Wind | | | √ | | | | | √ |

**Table A3.** LM-lag, LM-error, Robust LM-lag, and Robust LM-error tests for the OLS residuals for each season from 2020 to 2021.

| | **LM-Lag** | **LM-Error** | **Robust LM-Lag** | **Robust LM-Error** |
|---|---|---|---|---|
| 2020 MAM | 0.00 | 0.17 | 0.26 | 0.43 |
| 2020 JJA | 0.09 | 2.38 | 2.15 | 4.44 * |
| 2020 SON | 14.67 *** | 8.26 *** | 8.5 *** | 2.13 |
| 2020 DJF | 2.96 * | 2.81 * | 0.34 | 0.19 |
| 2021 MAM | 0.34 | 0.45 | 0.01 | 0.11 |
| 2021 JJA | 8.10 *** | 7.63 *** | 0.75 | 0.28 |
| 2021 SON | 9.90 *** | 1.95 | 10.99 *** | 3.04 * |
| 2021 DJF | 3.11 * | 0.11 | 5.74 ** | 2.74 * |

*, **, and *** denote the 10%, 5%, and 1% significant levels.

**Table A4.** Bandwidth based on MGWR in each season (The maximum bandwidth is 523,617.85 m).

|  |  | Bandwidth (Meters) |
|---|---|---|
| 2020 MAM | Intercept | 523,617.85 |
|  | Adult obesity | 523,617.85 |
|  | Access to exercise | 523,617.85 |
|  | Mental distress | 523,617.85 |
|  | Income inequality | 523,617.85 |
|  | % of persons in group quarters | 247,369.00 |
|  | % of persons below 18 years | 523,617.85 |
|  | %of native | 523,617.85 |
|  | % of rural | 523,617.85 |
|  | Air pollution | 523,617.85 |
|  | Humidity | 523,617.85 |
|  | Temperature | 523,617.85 |
| 2020 JJA | Intercept | 523,617.85 |
|  | Diabetes | 272,278.34 |
|  | Children in poverty | 523,617.85 |
|  | Income inequality | 272,278.34 |
|  | % of persons in group quarters | 523,617.85 |
|  | % of Hispanic | 523,617.85 |
|  | Humidity | 523,617.85 |
| 2020 SON | Intercept | 287,673.16 |
|  | Adult obesity | 523,617.85 |
|  | % of persons in group quarters | 247,369.00 |
|  | % of non-Hispanic | 312,582.50 |
|  | Wind | 352,886.67 |
| 2020 DJF | Intercept | 52,3617.85 |
|  | Mental distress | 523,617.85 |
|  | Diabetes | 352,886.67 |
|  | % of 18+ persons with at least one dose | 352,886.67 |
|  | % of 65 and older | 247,369.00 |
| 2021 MAM | Intercept | 523,617.85 |
|  | % of Asian | 523,617.85 |
|  | % of rural | 523,617.85 |
| 2021 JJA | Intercept | 262,763.82 |
|  | Poor health | 247,369.00 |
|  | Access to exercise | 523,617.85 |
|  | Diabetes | 523,617.85 |
|  | Mask | 418,100.18 |
|  | % of 65+ persons with at least one dose | 523,617.85 |
|  | Unemployment | 418,100.18 |
|  | % of non-Hispanic | 523,617.85 |
|  | Humidity | 458,404.34 |
| 2021 SON | Intercept | 523,617.85 |
|  | Diabetes | 523,617.85 |
|  | % of persons with at least one dose | 287,673.16 |
|  | Children in poverty | 523,617.85 |
|  | Inadequate facilities in house | 523,617.85 |
|  | % of persons below 18 years | 523,617.85 |
|  | % of Hispanic | 523,617.85 |
|  | % of non-Hispanic White | 523,617.85 |
|  | Humidity | 352,886.67 |
| 2021 DJF | Intercept | 247,369.00 |
|  | % of persons with at least one dose | 523,617.85 |
|  | % of 65 and older | 523,617.85 |
|  | % of Asian | 352,886.67 |
|  | % of native Hawaiian | 458,404.34 |
|  | Humidity | 458,404.34 |
|  | Wind | 523,617.85 |

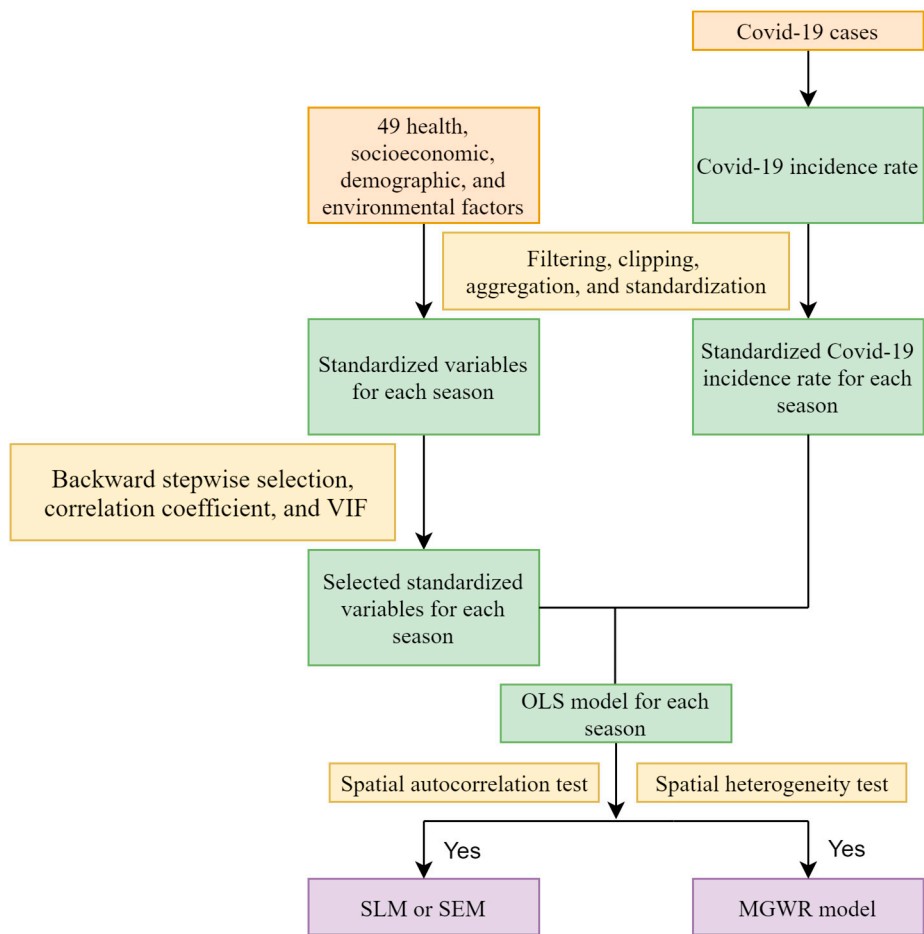

**Figure A1.** Flowchart of the method.

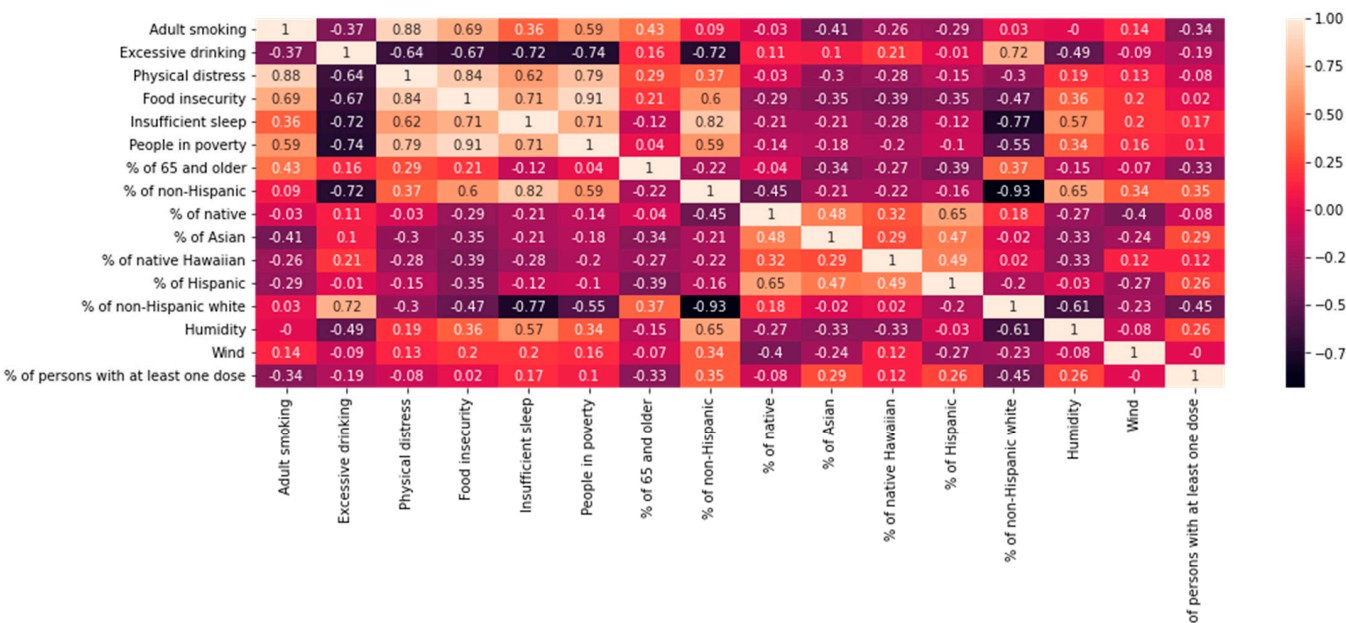

**Figure A2.** Correlation coefficients among different variables in 2021 DJF after backward selection.

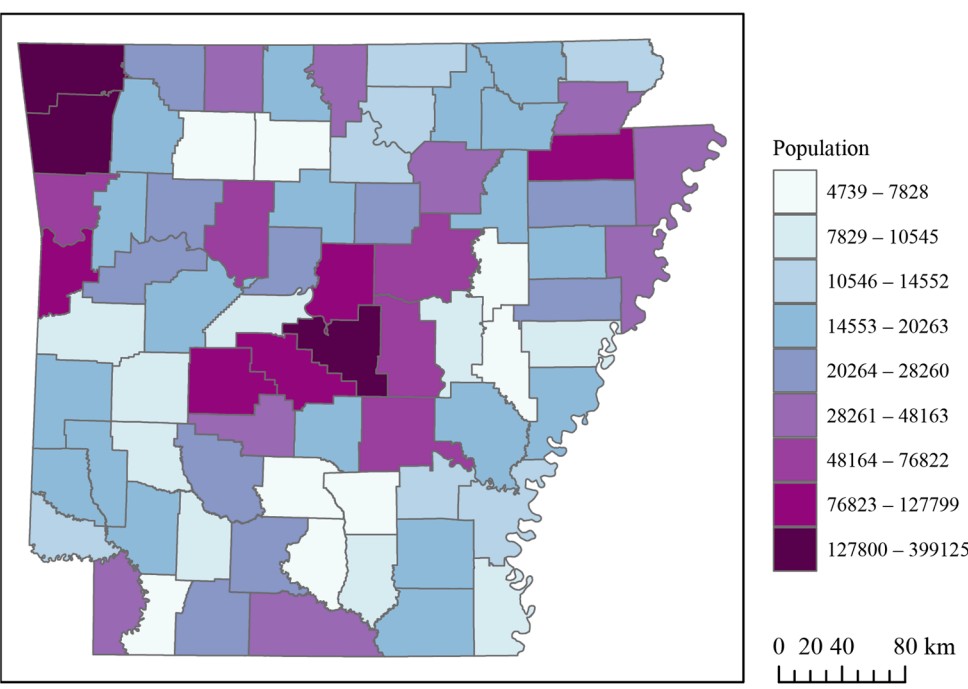

**Figure A3.** Population in 2020 in Arkansas.

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
