# Peer review of "Geospatial Modeling of Health, Socioeconomic, Demographic, and Environmental Factors with COVID-19 Incidence Rate in Arkansas, US"

_ijgi, doi:10.3390/ijgi12020045_

Round 1

Reviewer 1 Report

Major Comments:

- Novelty: This study do not contribute anything significant in the existing literature. Similar types of studies are conducted all over the world, including several states of USA. Also, the method is common. Your results and recommendations are nothing new and some of them do not have any practical implication. So, please, rethink and state the contribution of your study in a more meaningful way. 

-Methodology: I do not understand why do you use GWR? According to the assumptions of the OLS model, it cannot consider spatial heterogeneity and spatial autocorrelation. However, first you need to be sure that whether there is any heterogeneity in the relationships. From Table 2, it is clear that there is not spatially heterogeneous relationships in the models of 2020JJA, 2021MAM, 2021JJA, and 2021DJF. Also, you do not consider spatial scale or spatial autocorrelation. So, just saying we use GWR to explore spatial heterogeneity is not justified at all. First, we should review the spatial modeling framework and then you need to update your whole methodology. Check the following papers to know how to use spatial models. You need a massive change in the whole manuscript.

Comber, A., Brunsdon, C., Charlton, M., Dong, G., Harris, R., Lu, B., ... & Harris, P. (2022). A route map for successful applications of geographically weighted regression. Geographical Analysis.

Zafri, N. M., & Khan, A. (2022). A spatial regression modeling framework for examining relationships between the built environment and pedestrian crash occurrences at macroscopic level: a study in a developing country context. Geography and sustainability3(4), 312-324.

Minor Comments:

- Title: You should indicate that you explore association with COVID-19 incidence rate.

- Introduction: A large number of relevant literature are missing. Also, make a separate section for literature review. Add a table to present the summary of all the relevant literature.

- Materials and methods:

- You must provide summary statistics of all the independent variables.

- Describe section 2.3.1 in a more organized manner. 

- Mention the details of GWR model development procedure, i.e., bandwidth searching, spatial kernel, model type, optimization criterion, local collinearity assessment

- Results and discussion:

- Compare other parameters also in Table 2, i.e., AIC, AICc, CV, Adj R square

- Section 3.2 should be much more organized. Use sub-heading, bullet points to make it more readable. In current situation, it is extremely difficult to read and make comments.

- Put the result of the OLS models.

- Conclusions: Your recommendation section is poorly written. Improve it a lot. Also, state limitations of the study and future work scope in conclusion.

Author Response

Thanks, Reviewer for the very helpful comments. Please see my attached response letter.

Reviewer 2 Report

This study employed hot spot analysis and geographically weighted regression (GWR) to identify the spatiotemporal patterns of COVID-19 cases and quantify the associations of 49 variables at the county level in four seasons in Arkansas, US. These works are beneficial to reveal significant variables related to COVID-19 so that appropriate effort could be made correctly. However, the paper needs major improvement. Detailed comments are as follows:

Abstract

(1) Lines 25-27: Consistent with the existing finding that doing exercise has a negative association with COVID-19, this study also shows doing exercise may increase the chance of exposure to the virus, thus increasing the risk of COVID-19 infectionis not consistent with the conclusion in Line 335 “The access to exercise shows both negative and positive associations with COVID-19 during summer…”

Introduction

(1) Authors think that “it is necessary to explore the spatially heterogeneous impacts of different driving factors on COVID-19 spread across different time frames”, but the spatial heterogeneity is not considered in their aims (Lines 107-110).

Data and Methods

(1) Data preprocessing should be moved to the Data section.

(2) Python 3.8 libraries are just tools, and specific data preprocessing methods should be given. For example, the data standardization method can be Max-Min normalization but not Python 3.8 sklearn library.

(3) For the selection of health, socio-economic, demographic and environmental variables, quantitative results should be presented as a basis for selecting different variables for each season.

(4) Line 200: What does the unexplained x mean?

Results and Discussion

(1) According to Figure 2, COVID-19 cases mainly distributed in central, northwestern and northeastern regions in Arkansas, and there are many regions with relatively small numbers of cases. How to ensure the reliability of the GWR models established in these regions with few samples?

(2) Line 253: “This is likely attributed to the high population density in the regions.” This conclusion needs to be further proved by the spatial distribution of the population size or density in Arkansas.

(3) Line 281: “Interestingly, the relatively poor performances of GWR models dominate in 2021 MAM...”. If authors think GWR model for 2021MAM is related to the high infection in previous season, prove it! Take the immunity as a factor.

(4) The adjusted R-square value for GWR model in 2021 JJA is 0.65 as shown in Table 2, which is not the highest. However, Figure 4(f) shows that the local R-squared values for GWR of all regions are over 0.8 in 2021 JJA, which is the best performance in all seasons. How to explain this difference?

(5) In some spatial impact maps of some factors, e.g. Figure 5(d), 8(e), 10 (l), the numbers of counties with positive and negative GWR coefficients are almost equal. How to determine the positive or negative impact of this factor on COVID-19?

(6) The same factors at the same time have both positive and negative impacts on COVID-19 in different regions, and the same factors in the same region also have different impacts on COVID-19 at different times (even in the same season in different years). What causes these spatial and temporal differences?

(7) The title of this paper is to study the geospatial associations of health, socioeconomic, demographic, and environmental factors with COVID-19, but the Results and Discussion section has less spatial analysis, and only focus on the positive or negative correlation between these factors and COVID-19 during all seasons from 2020 to 2021.

Author Response

(The authors gave the same response as above.)

Reviewer 3 Report

This article discusses the relationship between multiple factors of geographical differences and the number of COVID-19 infections.

As a geographer, I can't judge the content of public health.

However, the GWR statistical method used in this paper is appropriate.

Thress suggestions:

1 Further increase the variable, because the flow of population across administrative regions will accelerate the increase of diseases.

2. The reasons for geographic differentiation in Figure 5 - Figure 12 generated by GWR need further explanation.

3 Line186 "Following a previous study by Song et al. [37], we selected the variables with VIF under 10." This may have serious multicollinearity.

Author Response

(The authors gave the same response as above.)

Author Response

(The authors gave the same response as above.)

Round 2

Reviewer 1 Report

I thank the authors to make huge improvement in the manuscript. However, I still have a major issue and some minor issues, which are as follows:

Major:

In methodology, you conduct Moran's I to detect spatial autocorrelation and based on which you make your decision regarding GWR modeling. However, GWR variant models are mainly used for exploring spatial heterogeneity in relationships. A well-fitted GWR may address the problem of spatial autocorrelation but which is not the primary purpose of the GWR models.

So, first, you need to explore whether there is any spatial heterogeneity in the developed OLS models. You may use Breusch-Pagan (BP) test, Koenker-Bassett (KB), White test, etc. to detect the presence of spatial heterogeneity in OLS models. If the result of these tests are significant, then using GWR/MGWR may improve the performance of the OLS models significantly. For example, I assume that BP, KB, and White tests result will be significant in the developed OLS models of 2020MAM, 2020SON, and 2021JJA. Therefore, developing MGWR improved their performance significantly. For other models, it may not be significant. Therefore, performance may not improve greatly for those models. By conducting these test, you can justify the rational behind using M/GWR model as well as you can say why the results for some model improve and not improve for some models. 

For spatial autocorrelation, you need to develop spatial lag model (SLM) and spatial error model (SEM). These models are primarily developed for addressing spatial autocorrelation in OLS model. For OLS model of 2020SON, 2020DJF, and 2021JJA spatial autocorrelation are present. After developing MGWR, spatial autocorrelation is still present in 2020SON model. So, I suggest to develop SEM and SLM to make the study more methodologically correct. See the previous two papers (mention in the first round) for further details.

Minor

1. Table 4 and Table 5: Table 4 is extremely under-utilized. In Table 4, you may only show the coefficients of the variables and their significance level for all models (e.g., OLS, MGWR, SLM, SEM). In Table 5, compare the model performance indicator.

2. Section 3.2.1 and Section 3.2.2 should be one section and you should write it in a comparative manner between the performance of different types of models.

3. It may be better if you sub-divide the Section 1 into a few sub-sections. 

Author Response

Thanks for the reviewer's helpful suggestions. Please find our attached response letter.

Reviewer 2 Report

Compared with the previous manuscript, the revised manuscript has made great improvements in method and organization. But there are still some works to be further improved.

 1. Results and Discussion section do not well answer the questions in the purpose of this paper, especially, question (2) (What are the spatial associations of different driving factors with COVID-19?). As my last comment of the previous manuscript, this paper mainly focuses on the positive or negative correlations between different factors and COVID-19, without describing and analyzing spatial differences in these correlations.

 2. It is worth thinking about why the correlations between some factors and COVID-19 are completely reversed at different times in the same region, such as access to exercise and % of rural, etc. The explanation given by the authors is a broad one, but there should be other specific reasons for the changes in the same area at different times.

 3. Several tables in the revised manuscript are too lengthy to read easily. It is suggested to change the expression.

Author Response

(The authors gave the same response as above.)

Reviewer 4 Report

I think the authors have addressed my comments and suggestions accurately. Accordingly, the manuscript has been quite improved and can be accepted in its current form.

Author Response

Thanks for the reviewer's support. Please find our attached response letter.
